# Somatostatin and Somatostatin Receptors in Tumour Biology

**DOI:** 10.3390/ijms25010436

**Published:** 2023-12-28

**Authors:** Ujendra Kumar

**Affiliations:** Faculty of Pharmaceutical Sciences, The University of British Columbia, Vancouver, BC V6T 1Z3, Canada; ujkumar@mail.ubc.ca; Tel.: +1-604-827-3660

**Keywords:** Somatostatin: somatostatin receptors, tumours, signaling

## Abstract

Somatostatin (SST), a growth hormone inhibitory peptide, is expressed in endocrine and non-endocrine tissues, immune cells and the central nervous system (CNS). Post-release from secretory or immune cells, the first most appreciated role that SST exhibits is the antiproliferative effect in target tissue that served as a potential therapeutic intervention in various tumours of different origins. The SST-mediated in vivo and/or in vitro antiproliferative effect in the tumour is considered direct via activation of five different somatostatin receptor subtypes (SSTR1-5), which are well expressed in most tumours and often more than one receptor in a single cell. Second, the indirect effect is associated with the regulation of growth factors. SSTR subtypes are crucial in tumour diagnosis and prognosis. In this review, with the recent development of new SST analogues and receptor-specific agonists with emerging functional consequences of signaling pathways are promising therapeutic avenues in tumours of different origins that are discussed.

## 1. Introduction

The tumour of different origins represents the most complicated and heterogeneous pathology that varies according to the tissue and site in an origin-specific manner, and therefore prognostic and diagnostic approaches are relatively different for each tumour. Despite significant progress, the rate of tumour-associated deaths is gradually increasing. The prominent research focus is on the discovery of gene therapy, and a majority of genes associated with the promotion and suppression of different type of cancer have been identified and characterized. However, the molecular mechanisms associated with the treatment failure and disease progression are still poorly understood. Previous studies revealed that 291 genes are associated with cancer-related malignancy, and the genes identified constitute 1% of the human genome. Amongst them, 90% resulted in somatic mutation in cancer, whereas 20% exhibited germ line mutation, and 10% were involved in both [1]. In addition to the gene therapy, several other molecules, including cell surface receptor proteins; hormones; mediator of downstream signaling pathways that account for tumour growth suppression, tumour proliferation and progression; and transcription factors that often lead to treatment failure, have been studied in cancer biology (Figure 1).

The metabolic events which are preferentially associated with certain enzymes have also been linked to tumour progression, as well as suppression. In this context, changes in energy metabolism and associated cell organelles can be critical determinants in tumour diagnosis. Moreover, the dysregulated aggressive cell proliferation critical for gradual tumour growth is also associated with the expression of cell cycle regulators, including cyclin dependent kinases (CDKs) [2]. In addition to tissue specificity, significant diversity in morphological, biochemical and molecular changes, as well as selective and specific modulation of signaling pathways, probably account for the poor diagnosis and prognosis of cancer. Importantly, one of the major challenges in treatment failure is the resistance to chemotherapy and the inability of drugs to reach the target. While tumour biology from the brain to peripheral tissues shares some commonalities, it also poses limitations and challenges that are critical in diagnosis and prognosis strategies. With the consistent increase in the number of cancer cases, early detection and adequate decision, as well as therapeutic preferences, might serve as the best approach for controlling tumour progression. An effective diagnosis of the tumour at an early stage is an opportunity for a better prognosis for most tumours of various origin. The increasing incidence of cancer cases and associated high mortality rate warrant the need to have a sensitive biological marker to detect, identify and characterize the tumour for immediate and effective treatment. These excellent observations that have emerged from extensive investigations have changed cancer research in regard to our understanding of pathophysiology and the elucidation of molecular determinants, providing new and effective treatment strategies. Several previous studies have shown the use of peptide hormone-based therapy in various types of tumour treatments. Somatostatin (SST), a growth hormone (GH)-inhibitory peptide, inhibits tumour cell proliferation and induces the prompt shrinkage of most tumours via binding to five different receptor subtypes that play an important role in a wide range of tumour treatment. Furthermore, for the key germline/somatic mutation, environmental factors; physiological abnormalities; and changes at the morphological, biochemical and molecular levels for the cancer of different origin are also associated with changes in SST and somatostatin receptors (SSTRs). Given the significant therapeutic role of SST in cancer, this review’s main focus is to briefly describe different types of tumours and elucidate the status and role of SST and its analogs in regulation of key molecular and cellular signaling pathways with specific interest in cancer biology as a potential therapeutic target.

## 2. Somatostatin: Beyond Inhibition of Growth Hormone

SST, also known as somatotroph release inhibiting factor, is a growth hormone inhibitory peptide which was first discovered in 1973 [3]. SST exists in two functionally active isoforms, namely SST14 and SST28, which are derived from a common precursor, preprosomatostatin. Both SST14 and SST28 are produced by SST-positive cells in various ratios in a region-specific manner and are found in central and peripheral tissues. The presence of SST is not restricted to the hypothalamus; it later was also detected in most body parts, e.g., central nervous system (CNS) and peripheral tissues with site-specific distribution [4,5,6]. Furthermore, cells displaying SST-like immunoreactivity have also been reported in selective species of invertebrates and plants, in addition to widespread distribution in all vertebrates [5]. The role of SST in the hypothalamus, pituitary and suppression of various hormones, including glucagon and insulin from the pancreas, inhibition of thyroid stimulating hormone (TSH)-stimulated release of triiodothyronine (T3) and thyroxine (T4) from the thyroid and regulation of gut motility, is well understood [5]. SST also serves as a neurotransmitter and neuromodulator in CNS. These functional diversities signify the role of SST in several CNS and peripheral pathophysiological conditions, including Alzheimer’s disease, Parkinson’s disease, Huntington’s disease, multiple sclerosis, diabetes mellitus, pain, obesity and satiety and inflammation, as well as neuropsychological disease such as schizophrenia, depression and anxiety, along serving a prominent role in cancer [5,6]. In addition to the inhibitory role on hormonal, cytokines and growth factor release, one of the most prominent and well-studied properties that SST governs is the antiproliferative effect that emerged from the study using octreotide (OCT) to prevent hormone hypersecretion in the intestine, pituitary and pancreas [7,8,9]. These observations further emphasized that such effects of SST are the inhibition of growth hormone and involve minimizing the tumour size with an antiproliferative effect via apoptosis. This effect has been much appreciated in tumour biology. SST exerts a distinct rate of tumour shrinkage in a tissue and tumour-specific manner through a well-known antiproliferative effect that may include cytotoxic and/or cytostatic effects. The antiproliferative effect of SST has been observed in normal dividing cells, including intestinal mucosal cells, activated lymphocytes, inflammatory cells and different experimental tumour models, as well as in cultured cells obtained from endocrine and epithelial tumours, including pituitary, thyroid, breast, prostate, colon, pancreas, lung and brain [5,10,11,12,13,14,15]. The first confirmative evidence supporting SST-analogs’ antiproliferative effect with clinical importance was derived from recruiting acromegalic patients with GH-oversecreting pituitary adenomas in a multicenter randomized trial [16]. OCT exerts an antineoplastic effect in somatotroph adenomas, and, when compared to untreated cases, it displayed a significant reduction in mean growth fraction. A nuclear protein expressed only in dividing cells, namely Ki-67, was reduced significantly in the GH-hypersecreting adenomas in patients pretreated with OCT compared to untreated controls [17]. Several previous trials have demonstrated the effects of SST-analog therapy on tumour shrinkage in acromegaly patients with a decrease in tumour volume [18,19]. Taken in consideration, three prominent roles of SST, namely anti-angiogenic, antiproliferation and pro-apoptotic, are linked to the suppression of different type of tumors.

## 3. Somatostatin Receptors: Therapeutic Implication and Clinical Significance

The different arrays of biological functions of SST are mediated via high-affinity binding to five different cell surface receptors, namely SST receptors 1-5 (SSTR1-5). SSTR subtypes exhibit 40–60% structural similarity and are encoded by five different non-allelic genes with different chromosomal presence. SSTR2 is further divided into two different isoforms, SSTR2A and SSTR2B. All five SSTR subtypes belong to the G-protein-coupled receptors (GPCRs) superfamily and have been well characterized pharmacologically. SSTR subtypes are widely expressed in different densities in various tissues in a receptor-specific manner in most species [20]. The different biological functions of SST are receptor-specific depending on the target tissue. SSTR subtypes couple to the inhibitory G protein (Gi) and inhibit enzyme adenylyl cyclase (AC) and suppressed formation of cyclic adenosine monophosphate (cAMP) and protein kinase A, as well as Ca^2+^ and K channel function, in a receptor-specific manner (Figure 2). In addition to the structural similarities, several previous studies have also shown overlapping functional properties and comparable regulation of signaling pathways.

Furthermore, studies have also revealed a distinct pattern of the SSTR subtypes’ distribution at the site of the tumour and regions in the tumour’s vicinity. However, post-SST agonist treatment, the activation of the second messenger and distinct physiological response of target cells further strengthen more than one receptor in a single cell. In cancer cells of different origins, the overexpression of SSTRs is often observed, with more than one subtype being expressed in a single cell. The presence of more than one receptor in a single cell postulate that SSTR subtypes might function in a heteromeric complex within the family (discussed later in detail). Taken in consideration, distinct levels of SST and its analogs’ effect depend on the presence of receptor subtypes at target. Given the importance of SSTR subtypes distribution in different tumours, the role of SSTR subtypes in regulating signalling pathways is discussed next as a potential therapeutic implication.

## 4. Somatostatin Receptors: Signaling Pathways and Cell Proliferation

The antiproliferative effects of SST, either induced directly or indirectly (Figure 3) in normal or pathologically driven target tissues, are the most prominent and effective antitumour activity. The antiproliferative effect is linked to the regulation of cell growth arrest and/or apoptosis with an implication of multiple complex but well-integrated signaling pathways. SSTR subtype-mediated activation or inhibition of the mitogen-activated protein kinases (MAPKs) has long been associated with cell growth inhibition in different cells [21,22,23,24]. SSTR subtypes, as discussed above, couple to Gi and inhibit enzyme AC, resulting in the suppressed formation of cAMP in most in vitro and in vivo experimental conditions. SSTR2-mediated inhibition of CHO-K1 cell proliferation is associated with activating extracellular-regulated kinase 1 and 2 (ERK1/2) and p38. [22,24]. Like SSTR2, activated SSTR1 is also involved in inhibiting cell proliferation via activation of ERK1/2 and p38 [21]. In contrast, SSTR5-mediated antiproliferative effects are attributed to the inhibition but not to the activation of MAPKs [25]. Furthermore, phospholipase C/inositol phospholipid/Ca^2+^ pathways have also been proposed in SSTR5-mediated antiproliferation [26], in addition to the induction of the retinoblastoma tumour suppressor protein (Rb) and p21 [27]. Surprisingly, in contrast to the well-established antiproliferative effect of SST, the stimulation of cell growth or proliferation has also been described in rare instances by MAPK activation via the activation of human SSTR4 [22].

Other factors associated with the direct effect of SSTR-induced antiproliferation are protein tyrosine phosphatases (PTPases) [28]. Previous studies using Ras-transformed NIH 3T3 cells have described how the PTPases in these cells are activated by SSTR2, SSTR3 and SSTR4 [29]. Florio et al. demonstrated SSTR1-induced cell growth inhibition in transfected CHO-K1 cells by activating SHP-2, which resulted in the MAPK pathway’s stimulation [21]. Comparable observations have also been reported for SSTR2 in non-mutated human gliomas [30]. In addition to SHP-2, the PTPase SHP-1 has also been demonstrated to be essential to SSTR2-mediated antiproliferation [31,32]. Activation of SHP-1 by SSTR2 inhibits the mitogenic response induced by insulin in CHO-K1 cells [33]. Both SHP-1 and SHP-2 were involved in the activation of ERK1/2, the induction of the cyclin-dependent kinase inhibitor p27^Kip1^ and the accumulation of hypophosphorylated Rb [23,34]. Moreover, the activation of a ternary complex involving SHP-1 and Janus kinase 2 (JAK2) was involved in SSTR2-mediated inhibition of intracrine fibroblast growth factor-induced cell proliferation [35].

In addition to the cytostatic effects, a cytotoxic effect leading to apoptosis has also been observed to the antiproliferative response upon treatment with SST. Three independent studies for the first time described apoptosis using AtT-20 and MCF-7 cells following treatment with OCT [36,37,38]. Furthermore, using breast cancer MCF-7 cells, SHP-1 dependency was revealed in SST-mediated apoptotic signaling via the activation of SSTR subtypes [36,39]. Since tumour cells often expressed more than one SSTR subtype, it was difficult to determine which subtype may be contributing to apoptosis. Sharma et al. later used CHO-K1 cells transfected individually with each SSTR subtype and found that apoptosis was uniquely triggered by human SSTR3 [40]. The authors further emphasized that apoptosis induced by the activation of SSTR3 is associated with the activation of p53, a tumour-suppressor protein, and pro-apoptotic protein Bax [40]. Contrary to Sharma et al.’s observations, p53-independent apoptosis via SSTR2 in HL-60, human pancreatic adenocarcinoma and human somatotroph tumour cells have also been reported [41,42,43]. In most neuroendocrine tumours, the expression of SSTR subtypes at the tumour site is associated with an SST-mediated antitumour role, supporting the role of SST as an endogenous inhibitor of cell proliferation [44]. Growing evidence identifies SST as an effective antitumour option which have an ability either to suppress the tumor promoting factors including certain kinases or cell survival pathways or via activation of anti-cell proliferation signaling. Taken together, the SST-mediated regulation of signal transduction pathways through the activation of different receptor subtypes depends on the inducer of cell proliferation in cells in a tissue-specific manner.

## 5. Somatostatin Receptors: Tumour Prediction and Treatment

Tumours of different origins represent the most complex pathology due to molecular, biochemical, physiological and morphological heterogeneity. Such complex tumour diversities at the molecular and cellular levels are linked to the failure in identifying the critical determinants of tumour indicators and/or specific markers that might account for unsuccessful single treatment. Multiple signaling pathways are associated with anti-apoptotic mechanisms, the suppression of tumor inducers that exert a cytotoxic and cytostatic effects and changes in different receptor proteins belonging to GPCRs and receptor tyrosine kinases (RTK) family involving in tumour suppression and progression, respectively. Previous studies have shown the negative effects of certain peptides on tumour growth; amongst them, SST, a growth hormone inhibitory peptide, is well known for its role in tumour suppression. At present, SST is a well-established potent therapeutic approach in treating tumours of different origins either directly via the activation of five different receptors exerting an inhibitory role on cell proliferation and the induction of apoptosis. SST and its analogs are also indirectly involved in regulating tumour suppression by inhibiting the secretion of growth factors or hormones that are actively linked to aggressive tumour growth in multiple tissues (Figure 3). In addition to the increased cell proliferation, several other factors also participate in an activation and progression of tumour. The next section of the review aims to describe certain key events that play a determinant role in tumour progression, including anti-apoptosis, angiogenesis and inflammation, which account for aggressive tumour growth and cell proliferation. Moreover, how SST and its analogs contribute to the pathology of cancer and interfere with tumour-promoting factors is discussed.

## 6. Somatostatin Receptors and Apoptosis: Suppression of Tumour Growth

Targeting apoptosis has served as one of the most effective therapeutic approaches in cancer. Oncogenes account for cell survival and blockade of cell death, including BCL2, elucidating the underlying mechanism of tumour growth [45,46,47]. The levels of anti- and pro-apoptotic BCL2 proteins monitored the release of cytochrome C from mitochondria that resulted in activation of different caspases, which are linked to apoptosis [46,48]. Furthermore, the regulation of transcription and phosphorylation of BCL2 proteins by CDKs and p53 also mediate apoptosis in a cell-specific manner [49]. Consistent with the notion of intrinsic and extrinsic factors involving in regulation of key molecular determinants of apoptosis, the role of peptide SST has also been appreciated. In addition to multiple well-articulated pathways, growth hormone-inhibiting peptide SST has been reported to induce apoptosis. The binding of the SST to five different receptors exerts an inhibitory role on tumour progression and cell proliferation. Studies also support that SST controls tumour invasion to an adjacent site. These tumour suppressor activities of SST are linked to cytotoxic and cytostatic effects, induction of CDKs and, importantly, apoptosis in normal and malignant cells via direct action of SSTR subtypes in a receptor-specific manner [50,51,52,53]. In support of SST-mediated apoptosis, the first evidence emerged in 1996 from a study revealed that OCT promotes apoptosis in CHO-K1 cells through SSTR3 by activating p53 and Bax [40]. Among the five different SSTR subtypes, SSTR2 and SSTR3 are the prominent receptors involved in the SST-mediated cytotoxic effect [14,40,54,55]. In HL-60 leukemia cells, SSTR2 exhibits apoptosis independent of p53. The overexpression of SSTR2 in MCF-7 breast cancer cells, which are devoid of estrogen receptor α (ERα), induced apoptosis and cell cycle arrest. The cells displayed a loss of epidermal growth factor (EGF)-mediated cell proliferation and epidermal growth factor receptor (EGFR) expression [54]. In non-transformed murine fibroblastic NIH3T3 cells, the expression of SSTR2 is known to potentiate tumour necrosis factor-α (TNFα)-mediated cytotoxicity [42,56]. SST by activating SSTR2 triggers apoptosis in several cell lines through the PTP-1C-mediated signaling pathway that results in activation of nuclear factor kappa β (NFK-β) and abrogation of c-Jun N-terminal kinase (JNK). These observations provide evidence in support of the role of SSTRs and death receptors’ crosstalk in modulating the pro-apoptotic signaling.

Most human pancreatic cancers lack SSTR2 expression [57] and exhibit apoptosis once transfected with SSTR2 [42]. Furthermore, SSTR2 enhanced TNFα-induced cytotoxic effect and TNFα-related apoptosis-inducing ligand (TRAIL) in caspase-3 and DNA repair protein PARP-1-dependent manner. SSTR2-mediated regulation of tumour growth is further supported by the receptor-induced activation of caspase-8 and post-transcriptional downregulation of antiapoptotic protein Bcl-2, as well as the upregulation of TNFα and TRAIL receptors.

The cytotoxic signaling events, including the robust time-dependent activation of p53 and enhanced levels of Bax in MCF-7 cells following treatment with OCT and proposed DNA fragmentation due to activated cation-insensitive acidic endonuclease to promote apoptosis, are well described [36]. OCT-mediated cell growth inhibition has also been reported in MCF-7 breast adenocarcinoma cells [38]. The cytotoxic effect of SSTR2 selective analog BIM23120 has been observed in human pituitary tumours. The pro-apoptotic effects attributed to increased activity of caspase-3 and PTP dependence without any other apoptosis-associated signaling molecules, including p53, Bcl-2 and Bax, were observed [43]. SSTR3-induced cytotoxicity has been reported in hepatocellular carcinoma cells upon treatment with OCT [55]. Both the SST- and SSTR-mediated modulation of immune function in lymphoid tissues have also been suggested [58]. OCT binding to human lymphocytes promotes apoptosis, and the underlying mechanism is the activation of caspase-3 and cleaved PARP-1, further providing evidence for the pro-apoptotic role of SSTRs in nontumoural lymphocytes.

## 7. Somatostatin Receptors and Angiogenesis

The diagnosis and prognosis strategies of several carcinomas have been improved significantly; however, tumour treatments are often disappointing due to the ability of the tumour to spread in adjacent and distant unaffected sites. The formation of new blood vessels, referred to as angiogenesis, is involved in several pathological conditions, including different types of tumours [59]. Moreover, the activation and proliferation of endothelial cells play a crucial role in pathogenesis of cancer, including breast cancer. The concept that most solid tumours are rich with blood vessels is a well-established and undisputed pathogenesis associated with aggressive tumour growth. New blood vessels might help in the clearance of certain waste products in circulation and provide nutrient and growth factors that support tumour proliferation. The formation of new blood vessels in tumourigenesis is a prime suspect in tumour metastatic which potentially guide metastatic dissemination, i.e., tumour growth at an additional site via tumour cells in circulation [60]. Most aggressive tumours often exhibit intense and dense vascular networks that serve as critical determinants of tumour nature and treatment preferences. Previous studies have shown that basic fibroblast growth factor (bFGF) and vascular endothelial growth factor (VEGF) are primarily involved in the regulation of tumour angiogenesis (Figure 4). Angiogenesis is not limited to one type of tumour; instead, it seems to be a classical feature of most tumours and related to tumour growth as a positive factor. It is interesting to note that tumours can stimulate angiostatin and endostatin, an angiogenesis inhibitor. These inhibitors have been used as an anticancer drug to suppress angiogenesis in primary tumour and tumour development at adjacent or distant site occurring through metastasis.

The perturbed balance between angiogenesis-promoting and -suppressing factors is believed to be critical in the formation of new blood vessels and an essential requirement of tumours for nutrients and oxygen. While angiogenesis is required for tumour metastasis, it also dictates whether the tumour will metastasize or not. Furthermore, the role of angiogenesis has also been supported in the regulation of the size of metastatic colony, as well as in tumour dormancy [60]. In well-articulated events of angiogenesis, several factors with potential promotion and suppression of tumours are well placed. In tumour aggressive growth and proliferation, the contribution of angiogenesis is enormous, and promoting factors include VEGF, bFGF, angiogenin, granulocyte colony-stimulating factor (GCSF), EGF hepatocyte growth factor, platelet-derived growth factor (PDGF) and cytokines [61,62,63]. To block angiogenesis-mediated tumour growth and metastasis, several natural angiogenic inhibitors, including thrombospondin; interferon; matrix metalloproteinase inhibitors; synthetic angiogenesis inhibitors such as protease inhibitors; anti-adhesive peptides; and pharmacologic inhibitors, including AGM 1470/TNP 470; and thalidomide, have been reported in addition to tumour derived inhibitor angiostatin and endostatin.

In addition to these prominent angiogenesis inhibitors, SST, which is widely distributed at a high density in different tumours and plays an antiproliferative role in tumours either directly or indirectly, has also been reported as a potential inhibitor of angiogenesis [64,65,66]. The presence of SSTR subtypes in the vasculature of different neoplasms and specifically the presence of SST in peritumoural veins might serve as an effective defense mechanism in angiogenesis and events associated with angiogenesis. Early studies using chicken chorioallantoic membrane model support the role of SST and its analogues, including OCT and RC-160 (bind to SSTR2), as an effective approach to blocking the development of new blood vessels and inhibition of neovascularization, i.e., angiogenesis [63,67,68]. The inhibition of angiogenesis seems to be an effective indirect mechanism of SST-mediated negative regulation of tumour growth. In addition to the direct action of SST in the regulation of tumour growth, the indirect action of SST analogs is involved in the regulation of growth hormone secretion and angiogenesis with a plausible mechanism involving the inhibition of VEGF. Patel et al. demonstrated that the anti-angiogenesis effect of OCT is blocked in the presence of increased calcium and cAMP and pertussis toxin (PTX) in contrast to no effect of protein kinase C (PKC) activation and inhibitors of tyrosine phosphatase [67]. The role of SST as an anti-angiogenesis peptide emerged from past studies describing the prevention of Kaposi’s sarcoma (KS) tumour xenografts’ growth in nude mice and inhibition of angiogenesis in a PTP-dependent manner [69]. The role of SSTR2 by using SST analogs that preferentially bind to SSTR2 with high affinity is known to inhibit angiogenesis in a dose-dependent manner [68]. Florio et al. further dissected the molecular mechanism and receptor subtypes in the prevention of angiogenesis via the SSTR3-mediated negative regulation of endothelial nitric oxide synthase (eNOS) and MAPK activities; by using the SSTR3 agonist, the effect was blocked in the presence of SSTR3 antagonist [44]. Consistent with the notion that neoangiogenesis is crucial in tumor growth, and migration and invasion are critical determinants of neoangiogenesis, the inhibition of migration and invasion might serve as an effective treatment in glioma. SST inhibits the migration in glioma cells by involving PI3-K and Rac activity, supporting the role of SST in regulation of invasion and metastases of glioma [70]. In parallel to these observations, Pola et al. reported the anti-migratory and anti-invasive role of SHSY-5Y cells that is associated with the inhibition of ERK1/2 and PI3-K signaling due to the inhibition of Rac [71].

Most importantly, the overexpression of SSTR5’s truncated form, sst5TMD4, in breast cancer model increases the pro-angiogenic factors, as well as the number of blood vessels, and is associated with the proliferation of and treatment response to SST analogs [72]. Furthermore, studies have also demonstrated a tumour-promoting effect in pituitary adenomas and thyroid cancer [73,74]. These observations further support the role of SST and SSTR subtypes in tumour proliferation. The invasion and metastasis are two well-connected processes that play a decisive role in tumour growth and failure in the treatment of cancer and are associated with angiogenesis. Therefore, the suppression of angiogenesis at an early stage once the tumour is diagnosed or the maintenance of tumour vascularization at balance might serve as a potential treatment that can be successfully achieved using SST and its analogues.

## 8. Inflammation, Pain and Cancer: Role of Somatostatin Receptors

Growing evidence from central and peripheral pathological conditions supports inflammation as a disease-promoting factor. Like many other pathological conditions, different types of tumours are associated with inflammation. The concept that inflammation is a contributing factor in cancer is well established and was first proposed by Virchow in 1863 [75,76]. The fact that tumour and the tumour-associated adjacent environment are composed of different types of inflammatory cells which are intimately associated with neoplastic events linked to tumour growth, migration and proliferation is well accepted and established [77]. Therefore, it seems that inflammation worsens tumour and leads to the failure of the tumour treatment. The mechanisms underlying inflammation in tumours are multiple, and inflamed tissue often exhibits aggressive growth due to the loss of balance between anti- and pro-inflammatory cytokines [77]. Moreover, the activation of pro-inflammatory signaling molecules and a lack of immunity are associated with aggressive tumor growth.

Furthermore, different leukocytes, as well as lymphocytes, present in tumour cells are able to release cytokines and chemokines that may trigger tumour progression via different mechanisms [77]. Moreover, a distinct pattern or intensity of damage due to chronic inflammation depends on the tissue-specific manner. Inflammation depending on contributing factors and its consequences may account for limited or aggressive tumour progression, as well as a significant impact on tumour growth suppression. Today, with significant progress in understanding tumour biology, the close interaction between tumour-promoting events, including inflammation and intrinsic immunity is well understood; however, the molecular mechanisms are still poorly understood. In addition to several well integrated factors, DNA damage prompted by inflammation is known prominent cause of tumour progression. The presence of SSTR subtypes on inflammatory cells including monocytes, macrophages, lymphocytes is the first indication for the role of SST in inflammation. Although controversy exist, all five SSTR subtypes are expressed in immune cells in receptor-specific manner at variable density [78]. The role of SST and its analogues as an efficient therapeutic alternative has also been proposed in inflammation and pain [79]. Previous studies have reported paracrine or endocrine mode of action that SST mediated in the regulation of inflammation [80]. Previously, using SSTR4 knock out mice studies have suggested that in the absence of SSTR4 mice are prone to inflammation and susceptible to pain is an indication that SSTR4 is the main receptor subtype that might involve in regulation of inflammation [81]. However, SSTR4 is not the main receptor in most tumours and future studies will be of great interest to attest the role of SSTR4 in cancer of different origin. SST and its analogs are involved in improving patients’ life span and comfort, in addition to the direct and indirect role that SST plays in tumour growth suppression. In this direction, the role of SST to enhance appetite and suppression of pain in cancer patients going through chemotherapy has also been reported.

## 9. Somatostatin, Somatostatin Receptors and Downstream Signaling Pathways: Mechanism of Cell Proliferation Inhibition

In addition to the changes in certain genes at the molecular levels, additional critical factors are associated with tumour progression and failure in treatment. Amongst them downstream signal transduction pathways are point of significance. Despite significant progress in understanding the role of signaling pathways in cancer, the molecular mechanisms associated with tumour suppression are still sparse. SST has shown selectivity in regulation of signaling molecules. SSTR subtypes are selective in modulation of signaling pathways in cells and tissues-specific manner. It is believed that pharmacological and physiological diversity exist in response to receptor-mediated functions. Multiple studies have shown the clinical significance of SSTR subtypes agonist in different type of tumours. Taken together, these observations attest to the significant role of SST in regulation of tumour growth or cell proliferation and the modulation of tumour promoting downstream signaling molecules; thus, it serves as potential therapeutic intervention in tumour biology. Accordingly, with existing concept and several studies supporting the well-established notion that SST inhibits cell proliferation, endocrine secretion, migration and inflammation via binding to five different receptor subtypes. SSTR subtypes are expressed in tumours of different origin at different levels and, once activated, are involved in regulating multiple downstream signaling pathways in a receptor- and tumour-specific manner. The purpose of the next section of this review is to discuss the role of SST and its analogs with receptor-specific biological functions associated in different type of tumours.

## 10. Somatostatin and Somatostatin Receptors in Brain Tumour

### Brain Tumour

A majority of tumours detected in the brain so far are the consequences of metastases. Although brain tumour is primary, the occurrence and prevalence of metastatic tumour are relatively higher than primary tumours in a ratio of 4:1 [82]. Due to different origins, brain cancer has not been associated with any specific symptom but seizures, headaches, confusion and nausea are frequently observed as generalized symptoms and focal symptoms including loss of language, unilateral weakness, and difficulty in walking [82]. Moreover, a wide range of variable symptoms, including the loss of cognitive function in severe cases; the type, size and growth rate; and the location of the tumour, are critical determinants of symptoms. Furthermore, the morphological distribution of metastases brain tumours is mostly detected in the cerebrum and brain stem, the brain regions associated with a different array of functions, including sight, hearing, taste, locomotion, thought and memory, balance and coordinated respiration, heartbeat, blood pressure and reflexes, respectively. Studies have also demonstrated that brain metastatic cells can have neuronal characteristics; however, the functional interaction between cancer cells and neurons is largely illusive [83,84]. Previous studies revealed that neuroblastoma is highly invasive tumour, and, once diagnosed, it tends to spread to bone, bone marrow and liver in 50% of the children with neuroblastoma [85].

In adults, the most common cancers that metastasize to the brain are shown in Figure 5. In large brain metastatic tumours are from lung and breast cancer [82]. One of the most frequently seen brain metastases amongst women emerged from breast cancer [86]. Although the prevalence of melanoma is relatively low, it displays high rate of incidence to metastasize to the brain. On the contrary, colorectal and prostate cancers display a lower rate of metastasizing to the brain. The cerebellum has been reported to be the target of gastrointestinal (GI) tumours; pelvic and renal cancers metastasize, whereas the posterior pituitary is often invaded by breast cancer. Although the molecular determinants are largely illusive, it is noteworthy that metastases to the brain are rare in children and account for 6% of total brain tumours.

Brain tumours constitute a major type of cancer, with approximately 20,000 deaths in North America each year. Epidemiological studies according to Canadian Cancer Statistics 2009 revealed that 2.1% cancer-related deaths in Canada are due to primary brain tumours. Of these, malignant astrocytic gliomas are the most common (50% of all gliomas) [87]. Glioblastoma multiforme is a highly invasive brain tumour due to rapid growth rate in comparison to other. The surgical procedure to excise these tumours, due to the potential risk of neurological abnormalities, is unsuccessful.

Furthermore, histopathological markers of well-developed brain tumours are accompanied by enhanced vascularization and necrosis, as well as increasing mitotic activity. In addition to mutations of cell division control protein 4 (cdc4) and mouse double minute 2 (mdm2) and deletions of p53, an aggressive brain tumour often exhibits overexpression of EGFR, VEGF and platelet-derived growth factor receptor. However, in the last four decades, very little progress has been made to improve the survival of brain tumour patients despite aggressive treatments because of the significant rate of recurrence [88,89]. Growing evidence also supports the formation of a synaptic connection between brain tumour and neurons that might play crucial role in tumor progression [90,91,92,93]. There are different types of tumors that exist in brain, including glioma, astrocytoma, glioblastoma, oligodendroglioma and oligodendrocytoma, medulloblastoma and ependymoma, and these are discussed briefly here.

## 11. Glioma

Glioma is one of the most frequently occurring malignant forms of primary brain tumour derived from non-neuronal glial cells (mature) or their precursor (poorly differentiated) cells. Gliomas constitute almost 80% of all brain tumours, and this is the leading cause of death in patients with tumour growth, both children and adults. Glioma is the brain tumour that displays significant variations that are linked with multiple factors, including age and gender, site of location, degree of invasiveness, ability to progress and grow, and distinct morphology. Neuronal cells constitute major part of glioma micro-environment and neuronal activity exert a crucial role in aggressive tumour progression and growth by releasing growth factors [92]. Although glioma’s invasion of other tissues is very rare, tumour growth in other brain regions and the spinal cord has been described. Experimental mouse models of gliomas have shown valuable molecular, biochemical and morphological details. Because there is a lack of effective therapeutic options and those that are offered are limited, patients with gliomas often die in a very short time post-diagnosis. The therapeutic option available includes surgical removal, radiotherapy, chemotherapy and immunotherapy [94,95]. However, therapeutic intervention directly to the tumor site or tumor microenvironmental factors and neuronal activity is the question of debate for future research. Tumour infiltration and proliferation have been associated with the ability of glioma cells to form tumoural microtubules [96]. Glioma with solid and isolated growth patterns, with newly formed blood vessels with less parenchyma or intact brain parenchyma, respectively, have been identified.

## 12. Glioblastoma

Glioblastoma is the most common, aggressive and lethal primary brain tumour in adults with an impaired BBB and represents 16% of all brain tumours and 54% of gliomas [97,98]. The significant variations in glioblastoma composed of poorly differentiated neoplastic cells in a brain region-specific manner are of considerable challenges in analysis of glioblastoma [99]. Amongst the most brain tumours, glioblastoma is highly invasive and fast growing, with dense vascularization at the tumour site associated with uncontrolled aggressive tumour growth. The aggressive tumour growth is linked to the critical contribution of EGFRs, which are well expressed in most glioblastoma. In large cases of glioblastoma, changes in signalling pathways, including RTK-RAS and PI3K, are frequently seen; furthermore, cell proliferation and tumour growth are linked to the mutation in these signalling molecules [100]. The diagnosis and prognosis of glioblastoma are difficult due to aggressive cellular proliferation and growth. Glioblastoma, with its with highly invasive nature and its complex and heterogenous pathology and location in the CNS, is resistant to most available treatment.

## 13. Meningioma

Meningiomas originated from arachnoid cap cells, which constitute 30–35% of all intracranial tumours and are believed to be the most common brain tumours in the adult population [101]. According to the WHO, meningiomas are in grade I-III, with large number (90%) in grade I; high grade (WHO grade II) and grade III anaplastic are the aggressive types and tend towards local invasiveness, with an incidence of recurrence and high rate of mortality. In the case of a large or symptomatic tumour, surgery is the preference of treatment. Chemotherapy is not well established for the treatment of meningioma. Previous studies support the recurrence of highly aggressive meningioma [102,103,104,105]. The gradual progression of tumour growth resulted in shrinkage of brain regions in the vicinity of the tumour and resulted in neurological issues, i.e., seizure and neuropsychological disturbances [106,107].

## 14. Medulloblastoma

Medulloblastoma is the most common and aggressive embryonal brain tumour (primarily in cerebellum) seen in children and accounts for 15–30% of all childhood brain tumour cases. Previous studies support that medulloblastoma is composed of clinically and molecularly different tumours [108,109,110]. Medulloblastoma in adulthood is confined to the cerebellar hemisphere and accounts for 1–3% of primary brain tumours [111,112,113]. Medulloblastoma is derived from premature neuronal progenitor cells and is frequently in cerebellum. The large numbers of medulloblastoma originated due to Hedgehog pathway activation in granule neuron precursor cells.

In contrast, medulloblastoma with mutation in the WNT pathway originates from the brain stem. It is believed that, at the advance stage, neoplasia invades other brain regions through cerebrospinal fluid (CSF) [111]. Past studies have demonstrated that variation in biological differences in medulloblastoma are linked to morphological characteristics feature of tumours [114]. The prominent genetic risk factors linked to this neoplasia include Hedgehog pathway, the wingless (Wnt) and c-myc [115,116]. In medulloblastoma pathogenesis, sonic hedgehog signaling via binding of SHH to it receptor exert a determinant role [114,117,118,119,120]. Al-Shardah et al. described the molecular details regarding different locations, radionuclide imaging and clinical descriptions of medulloblastoma [121]. Medulloblastoma is a very heterogenous tumour in nature, and the pathological events associated with tumour progression are not well understood. However, highly invasive tumour cells in periphery of solid tumour tissue in medulloblastoma have been observed [122].

## 15. Somatostatin Receptors Expression and Brain Tumours

In addition to the changes in many growth factors and neuropeptides, several other cell surface receptor proteins are expressed in brain and the imbalances in proportion of these proteins associated with the brain tumour progression. SST and its SSTR subtypes are expressed at higher levels in various malignancies, including brain tumours [20,123]. High densities of specific SSTRs and often multiple receptors in same cells have been reported in various forms of brain tumours, including, gliomas, medulloblastomas, astrocytoma and neuroblastomas, as well as meningioma [122,124,125,126,127,128,129]. Receptor binding studies have shown the expression of SSTRs in various categorized (WHO grade I–II) gliomas. Amongst all five SSTR subtypes, SSTR2 is the receptor subtype that was found to be overexpressed in different WHO-grade solid human gliomas, as well as in human glioma cell lines [130,131]. We previously showed the mRNA expression of all five SSTR subtypes in glioma and meningioma samples using Northern blot analysis and RT-PCR techniques and described that SSTR2 mRNA was the most abundantly expressed in both meningiomas and glioma samples [123]. However, in this study, no correlation could be established between the mRNA expression and site or histology of the tumour [123]. Although SSTR 2 is the predominant subtype expressed in these brain malignancies, Lamszus et al. reported the expression of SSTR1 and SSTR3 mRNA levels in 7–9 cell lines derived from human gliomas, as well as in a few cases of glioblastomas; however, SSTR4 and SSTR5 mRNA levels were rarely seen [132].

The expression of SSTRs in glioblastoma, however, remains controversial, as Luyken et al. demonstrated the expression of SSTRs with an SST-gold conjugate [133]. In contrast, Reubi et al. failed to detect the SSTRs in glioblastomas [124]. In meningioma, all five receptors are well expressed in a receptor-specific manner with SSTR2 as a major receptor subtype; however, the expression of SSTR5 was relatively higher in benign than malignant meningioma [134]. These observations also revealed that SSTR expression is not related to the age, sex, tumour site and recurrence. Also, high expression of SSTR2 and -3 has been shown in meningioma after partial resection with tumour recurrence [134]. Previous studies also revealed the role of hormonal factors as a critical determinant for tumour growth and described the expression of estrogen, progesterone receptors and androgen [135,136,137].

The development and growth in a high number of glioblastoma cases are under the influence of EGFR-mediated signaling pathways. In contrast, activated SSTRs suppressed tumour growth and also block EGF-induced cell proliferation and ErbBs complex formation in breast cancer cells [2,36,38,52,54,72,138]. These observations lead to the assumption that targeting SSTR subtypes in glioblastoma might serve as an effective treatment approach. Direct support to such speculation emerged from previous studies describing the role of SST-, OCT- and SSTR2-specific non-peptide agonist in regulation of cAMP, SHP2, MAPK using human U343 glioma cells that resulted in dephosphorylation of EGFR and PDGFR linked to the inhibition of ERK1/2 phosphorylation [30]. Furthermore, the EGF-mediated stimulation of AP-1 complex at the molecular and protein level is suppressed in presence of SST and SSTR2 selective agonist [30]. In addition, antiproliferative effect of SST in glioma cells has also been reported through increasing DEP-1/PTP activity and inhibition of ERK1/2 activity [139]. Moreover, the loss of phosphatase and tensin homolog (PTEN) and tumour resistance to tyrosine kinase inhibitors, glioblastoma can also be averted in combination with SSTR agonist.

In medulloblastoma, by using Octreoscan scintigraphy, a high expression of SSTR subtypes 2 and 5 has been reported in vivo [140,141,142]. Immunohistochemical mapping of SSTR2 revealed a differential distribution pattern of SSTR2 in xenograft tumours which are confined to the cell surface in contrast to the cytoplasmic and nucleus in cultured medulloblastoma cells that possibly associated with OCT-mediated effect [143]. Furthermore, comparative distribution of SSTR subtypes mRNA has also been made in glioma and medulloblastoma with comparable expression to control in medulloblastoma with high SSTR1 and least expression of SSTR5, whereas, in glioblastoma, expression was different, and tumors displayed high expression of SSTR3 and minimum expression of SSTR5 [122]. High expression of SSTR2 leads to SST and analogs scintigraphy. Furthermore, studies have also demonstrated the association of SSTR subtypes’ expression whether the brain tumour is differentiated or undifferentiated. SSTR subtypes are present in differentiated brain tumours, meningiomas, low-grade astrocytoma and oligoastrocytoma compared to undifferentiated glioblastoma, which normally are negative to SSTR subtypes [129,132,144,145]. Taken into consideration, these observations suggest that the presence of SSTR subtypes in differentiated brain tumours might serve as an experimental tool in tumour prognosis.

## 16. Brain Tumours and Therapeutic Alternatives

The tumour of brain origin is considered difficult to treat. First, due to multiple complex functions; second, failure of drugs to reach the target due to blood–brain barrier (BBB) impermeability to most therapeutic drugs; and third, resistance to radiation and chemotherapy [99]. The effective and standard treatment for brain tumours is surgical resection but depends on the patients’ conditions. Moreover, radiotherapy is frequently used in cases where surgery is not a possible option. The heterogeneous nature of brain cancer often resulted in resistance to single targeted therapy. Like many other neurological diseases, one of the major limitations in treating brain tumours is the permeability of many agents across the blood–brain barrier that excludes the CNS. Also, there are limited drug to choose from for oral administration, including temozolomide, procarbazine and lomustine, or to administer intravenously (e.g., vincristine, cisplatin, carmustine (BCNU) and carboplatin). Temozolomide is 100% bioavailable and effective in crossing the BBB and is also well tolerated; however, some patients develop resistance to the drugs. It is believed that several drugs used in chemotherapy do not cross the BBB due to being hydrophilic in nature and large in size. In addition, some chemotherapeutic drugs, including cisplatin, have been used in combination with mannitol for the treatment of brain tumours.

Furthermore, OCT has shown promise to inhibit the proliferation of some brain tumours, including gliomas and medulloblastomas [88,146]. Contrary to these observations, Mawrin et al. reported no correlation between SSTR expression and tumour proliferation and whether SSTR subtypes present or absent in tumours have no role on survival [147]. Recently, in children with medulloblastoma, cerebrospinal fluid-derived cell-free DNA was identified as a biomarker of measurable residual disease [148].

## 17. Role of Somatostatin and Somatostatin Receptors in Brain Cancer

Understanding the morphological, biochemical and molecular diversity of brain tumours is a critical determinant of effective therapeutic strategies. Furthermore, the presence of other neuropathological conditions, including neurological diseases, also pose significant interference in neuroimaging and other associated examinations and make brain tumour diagnosis more difficult. In addition to EGFR, several receptor proteins including GPCRs are also well expressed or often overexpressed in various types of tumours and serve as a potential therapeutic approach. Amongst them SSTR subtypes prominent member of GPCR family are highly expressed in most tumours of peripheral tissue and have shown a crucial role in tumorigenesis. Still, their expression in brain tumours is limited and controversial. As discussed above, SSTR subtypes are variably expressed in different brain tumours. They are related to the diagnosis process, and clinical benefits with the use of SST analogous are still disputed. The mapping of SSTRs in various brain tumours led to the application of radiolabeled imaging and targeting of these tumours using radiolabeled SST analogs in gliomas and meningioma [7,129,133]. High-grade brain tumours are often associated with suppressed cAMP expression levels [149,150,151]. This inverse relation is further supported with observation describing inhibition of cell proliferation and induction of apoptosis with increased cAMP expression [149,151,152].

Contrary to these observations, despite the inhibition of cAMP in the presence of SST and its analogs, the inhibition of cell proliferation and induction of apoptosis is well established. Although mechanism involved are poorly understood, previous studies using porcine primary culture have shown stimulatory effect of SST on cAMP at both low and high doses [153]. The known antiproliferative effect of SST in pancreatic, pituitary and breast tumours and mapping of SSTRs in meningioma and gliomas indicate an effective therapeutic value of SST in the treatment of these tumours. Growing evidence supports the idea that SSTR2, in comparison to other meningioma markers, is more sensitive in the diagnosis of meningioma [154,155,156]. The expression of SSTR2 in meningioma might serve as an effective diagnostic and treatment strategies [106]. In most brain tumours, meningioma is a type of tumour that displayed relatively high expression of SSTR subtypes and attested significant role in meningioma when compared to other brain tumours [157,158,159,160]. Schulz et al., using receptor-specific antibodies, demonstrated strong expression of SSTR subtypes, specifically SSTR2 and further emphasized that SSTR2 agonist is a possible effective therapy [161]. Despite some limitations, OCT has been implicated as a successful treatment regime of meningioma [161,162,163,164]. Studies have shown the use of ^68^Ga-DOTATATE, which recognizes SSTR2 and is being used clinically to evaluate or identify meningioma. SSTR specific newly nonpeptide agonists hold some promise in treating brain tumours [165].

In contrast to tumour suppressive effect, the growth-promoting effect of SST analogs on cultured human meningioma cells is also evidenced [166,167]. Graillon et al. described that OCT holds promising antiproliferative activity in meningioma in vitro with no induction of apoptosis implicating these results with suppression of tumour growth rather than tumour shrinkage in vivo [168]. These results further supported the inhibition of AKT activation in addition to activation of tyrosine phosphatase without any role on ERK1/2. Moreover, the expression level of SSTR2 at protein but not at mRNA level is associated with a better response of OCT in cell viability [168]. Taken together, widespread distribution of SSTR2 in meningioma implicates SSTR2 radiolabeled ligand in the diagnosis of meningioma. Contrary to most studies supporting SSTR2, no correlation of SSTR2 has also been reported with WHO grade meningioma [165]. Furthermore, the presence of SSTR2 in partially resected meningioma has also been proposed possible cause or risk factor for tumour recurrence [134]. SSTR2 inhibits the cAMP levels in glioma cells and, in addition to the modulation of potassium channels leading to the hyperpolarization in glioma cells of human origin [131]. Furthermore, SST is also known to antagonize the effect of epidermal growth factors and stimulates tyrosine phosphates such as SH-PTP1 and SH-PTP2, which consequently dephosphorylates the EGF receptors in breast and pancreatic cancer, thus exerting an antiproliferative effect [26,169,170]. This signaling mechanism of SST could be a plausible way of exerting an antiproliferative effect in brain tumours.

Surgical resection is the well-established and standard treatment for patients with medulloblastoma and later followed by radiotherapy [111]. Despite lacking suitable chemotherapy, a therapeutic approach in combination has been employed in brain tumor treatment. Stefanović et al., for the first time, described the use of SST and its analogs in the treatment of medulloblastoma in children [171]. Previous studies have shown encouraging results in the treatment of medulloblastoma following intrathecal administration of radiolabeled SST analogs despite controversial observations with non-radiolabeled OCT [143,146]. However, OCT at a relatively high concentration inhibits cell proliferation in vitro but failed to show any effect in combination with cytostatic drugs [143]. Galvis et al. claim that OCT is a suitable therapeutic intervention for medulloblastoma [112]. The molecular mechanism for SSTR2-mediated effect in in vitro, as well as in animal, studies includes the regulation of adenosine triphosphate (ATP) and nucleic acid synthesis that resulted in the suppression of metabolic activity in addition to the modulation of MAPK pathway and also might inhibit cell proliferation (mitosis) and activation of apoptosis [112,172]. These observations and the concept that SSTRs function as an effective suppressor of tumour promoting cellular signal and inducer of apoptosis, SSTRs may serve not only as a marker but also as a therapeutic alternative in brain tumours.

## 18. Primary Central Nervous System Lymphoma

In contrast to brain tumors as discussed above, primary central nervous system lymphoma (PCNSL) is the type of brain tumor which is not associated with SST, neither for diagnosis nor prognosis. SSTR scintigraphy is frequently used to visualize the tumors positive to SST but not PCNSL [173]. Brain regions affected with PCNSL include basal ganglia, thalamus, periventricular white matter [174,175]. It is also interesting to note that brain regions, despite of good expression of SST and SSTR subtypes, exhibit no association with PCNSL pathogenesis. This is a rare type of brain tumour that is often seen in older populations and often seen in immunocompetent and immunocompromised patients [176] and human immunodeficiency virus infection is a risk factor in the development of PCNSL with poor prognosis [177]. In CNS, lesions seen close to the ventricle in lymphoma and affected area with lymphoma, including the spinal cord, cranial nerves, brain parenchyma, meninges and intraocular structures and brain biopsy, are well-recognized as a diagnosis of CNS lymphoma. In primary CNS, lymphoma hypothalamic–pituitary lesion has also been reported [178]. With significant progress in understanding the molecular complexity of different brain tumours, we still face challenges in developing potential therapeutic interventions.

## 19. Blood–Brain Barrier Critical Determinant of Brain Tumour Treatment

Therapeutic choices in the treatment of brain tumours are limited due to the failure of drugs to cross the blood–brain barrier, and many of the drugs that are easy to target at the tumour site in peripheral tissue are unable to reach the brain due to their increased size and being hydrophilic in nature, unless the BBB is disrupted. The brain is isolated from the rest of the body by the well-organized BBB, which has the ability to block certain substances and cells from entering the brain, which is larger in size and hydrophilic in nature. Most chemotherapeutic drugs available for tumour treatment fail to cross the BBB. An intact BBB poses a potential threat in the successful treatment of brain tumours. Brain tumours with an intact BBB include glioma of low grade in contrast to brain tumours originating from intracranial tissues devoid of a BBB [179].

## 20. Somatostatin Receptors and Breast Cancer

### 20.1. Breast Cancer

Breast cancer is the leading cause of death among women aged 40–50 years across the globe and is characterized as one of the most complex and common forms of cancer diagnosed in women. The heterogeneity of this disorder, involving serious changes in the normal functioning of the mammary epithelial cells, leads to a genetically altered cell phenotype [180]. It is believed that one-third of cancer is hormone-dependent [181], and several ovarian hormones, such as progesterone (P) and estrogen (E), as well as growth factors, including EGF and insulin-like growth factor (IGF), which are normally secreted by the mammary cells and aberrantly secreted by the carcinogenic cells, lead to a poor prognosis of the disorder [180]. In addition to hormonal disturbances, tumour-promoting signalling pathways and enhanced anti-apoptotic machinery, as well as elevated circulating cholesterol, are reported to be associated with breast cancer. Furthermore, increased recurrence-free survival is seen in breast cancer patients with the use of statin, a known cholesterol-lowering drug [182,183,184]. Baek et al. further emphasized that 27-hydroxycholesterol enhanced the secretion of extracellular vesicles [182]. The cholesterol metabolite 27-hydroxycholesterol promoted tumour proliferation in ER-positive tumours and induced the suppression of anticancer immune response and metastasis via liver X receptors in myeloid cells [185,186,187,188,189]. The complex heterogeneous pathology of breast cancer is possibly the leading cause of poor diagnostic and effective treatment strategies. Although the inherited mutation in BRCA1 and -2 known tumour suppressor genes is a well-defined predictor of breast cancer, a strong correlation between the presence of SSTR subtypes and breast cancer might also predict some of the therapeutic approaches. This section first describes the expression of SSTR subtypes in breast cancer and then the use of SST and its analogs as a potential therapeutic alternative in treatment of breast cancer.

### 20.2. Expression of SST and SSTR Subtypes in Breast Cancer

SST and SSTRs are well expressed in breast cancer cells and autopsy breast tumor tissues. SST and SSTRs have been studied for their antiproliferative effect in various types of tumours, including breast tumours. Several studies have characterized the expression of SSTRs in neoplastic and physiological tissues. The presence of SST in the mammary gland of lactating rats, as well as expressing proprotein convertase PC1, which is associated with the processing of SST has been reported [190]. SST-like immunoreactivity has been demonstrated in approximately 30% of the breast tumour tissues and breast cancer cell lines. It is believed that there are direct and indirect mechanisms of action of SST on breast tumour cells (Figure 3). The direct biological actions of SST or its analogs are via the binding to different SSTR subtypes involving inhibition of cell proliferation and/or induction of apoptosis. The indirect effect of SST may be due to the inhibition of the growth factors such as IGF-1 gene expression, which indirectly inhibits the levels of IGF-1 overexpressed during tumour development [138,191]. Previous binding studies have shown that 15–66% of primary breast tumours are positive for SSTRs, whereas 75% were positive when imaged in vivo, using [111In-DTPA-DPhe1]-octreotide scintigraphy [192,193]. According to a study carried out by Pfeiffer et al., SSTR2 and SSTR5 were the predominant subtypes expressed in primary breast tumours [194]. The expression of SSTR subtypes at the levels of mRNA and protein for SSTR1 and -2 has also been reported with the effect of Tamoxifen and estradiol in breast cancer cells [195]. Several other studies have also reported SSTR2 to be the most abundant SSTR subtype expressed in breast tumours [192,196,197,198]. SSTR2 expression in breast tumour tissue was also found to be ubiquitous [196]. Viki C-Topic et al., in-breast carcinoma, illustrated the incidence of SSTR subtype transcripts in a breast tissue sample. According to their study, SSTR2 transcript was predominantly expressed in all samples, followed by SSTR1, SSTR3 and SSTR4 [199]. Moreover, at least one other subtype of SSTR transcript, mostly SSTR1, was detected along with SSTR2 transcript in 96% of the breast tissues examined [199]. Furthermore, the expression of both the mRNA and protein levels of all SSTR subtypes was shown in a cumulative study in ductal NOS breast tumours [192]. Additionally, it was suggested that the SSTRs are variably distributed in the same breast tumour tissue and in the surrounding tumour regions [192]. This study from >90 cases of breast cancer led us to establish a correlation between tumour growth, hormonal levels and morphological changes with a distributional pattern of SSTR subtypes. SSTR1 and -4 were correlated with estrogen receptor (ER), whereas SSTR2 was correlated with progesterone receptor (PR), in addition to ER [192]. The expression of SSTRs was found to correlate with the tumour grade and the levels of ER and PR, which contraindicated the study carried out by Vikic-Topic et al., showing no correlation between the SSTR transcripts and patient age, tumour grade and ER, as well as PR [199]. SSTR1 and -4 were correlated with ER, whereas SSTR2 was correlated with PR in addition to ER [192,199]. Zou et al. recently demonstrated that all five SSTR subtypes are expressed in breast cancer tissues with a relatively high expression of SSTR1(90%) and SSTR4 (71.3%) and suggested inhibition of cancer cell proliferation independent of patient age and stage of cancer [200]. Studies have also shown the expression SSTR2 and SSTR5 in neuroendocrine breast cancer. An interesting observation emerged from Castano’s laboratory from pituitary adenoma: a new truncated variant of SSTR5, namely sst5TMD4, was identified [73]. The presence of this truncated receptor is associated with proliferation and poor response to SST analogs. sst5TMD4 overexpression has also been reported in thyroid cancers. However, the presence of sst5TMD4 in breast cancer is associated with poor prognosis, and sst5TMD4 expression in breast tumour cells is linked to increased malignant feature [72,73,74]. The overexpression of sst5TMD4-induced angiogenesis and pro-angiogenesis mechanism resulting in tumour progression has been proposed [72]. Thus, the pattern of SSTR subtypes’ distribution in breast cancer is heterogenous and inversely related to the expression of EGFRs, and SSTR subtypes are often seen in breast tumours with an expression of estrogen and progesterone.

### 20.3. Role of SST and SSTR in Breast Cancer: Possible Treatment Choice

The successful treatment of any tumour, including breast cancer, depends on detection at an early stage. Growing evidence and consistent observations accumulated from different studies support an undisputed direct and indirect role of SST and its analogues on the negative regulation of cell proliferation. The functional significance of inhibition of hormonal secretion and growth factors, including the inhibition of IGF1 expression and serum level, was further explored as effective adjuvant therapy for tamoxifen treatment in breast cancer; however, later studies failed to support the idea of there being any beneficial effect of SST in combination with tamoxifen [201,202,203]. Prior to delineating the direct and indirect effect of SST and its analogues in the suppression of cell proliferation, induction of apoptosis and regulation of tumour-promoting signaling pathways, several radiolabels were used for diagnosis purposes. Several previous studies using in vitro, in vivo and animal models have described antiproliferative and reduced tumour growth following treatment with SST and its analogues. Over the years, various SST analogues (Table 1) have been used as antiproliferative agents in the treatment of breast cancer. Unlike SST with a short plasma half-life of 3 min, it has been shown that SST analogs have better efficacy, a better therapeutic index and are free from major side effects [204,205,206]. Earlier reports by Setyono-Han et al. showed the inhibitory effects of Sandostatin, an analog of SST, on the proliferation of human breast cancer cells [207]. In a concentration- and time-dependent study, Sandostatin at 10 nM optimally inhibited the growth of the MCF-cells. SST analogs had an antagonizing effect on estradiol and growth hormones in vitro in these mammary tumour cells. Setyono-Han et al. observed similar effects with SST, suggesting that SST and SST analogs directly act as potential antiproliferative agents on human breast cancer cells [207]. Vapreotide, another analogue of SST, was evaluated, and it was found that prolonged treatment with vapreotide was well tolerated in the case of pretreated metastatic patients without any side effects [208]. Moreover, the levels of IGF-1 were diminished and remained stably reduced the entire length of the treatment. This study suggested the beneficial effect of Vapreotide as a continuous treatment suppressing the levels of growth factors, usually hypersecretion in breast cancer. A similar study by Canobbio et al. indicated that the SST analog Lanreotide also suppressed the levels of IGF-1 in postmenopausal breast cancer patients previously untreated for the tumour [209].

OCT is the most studied SST analog that has been used for a long time in certain tumours treatment, including breast cancer. In combination with Tamoxifen used for the treatment of breast cancer, OCT showed enhanced antiproliferative effects in DMBA i-duced rat mammary carcinoma. OCT also effectively increased the antineoplastic effect of ovariectomy in these rat models [210]. Sharma et al. demonstrated that SST exert a cytotoxic effect on MCF-7 cells in a receptor-specific manner and found that SSTR3 uniquely participates in the induction of apoptosis [40]. Furthermore, it was reported that in human breast cancer MCF-7 cells, SST analog OCT triggered apoptosis by stimulation of a tumour suppressor protein, wild-type 53 and Bax, suggesting the potential use of SST analogs in the treatment of breast cancer cells expressing SSTRs, as well as wild-type p53 [36]. In a novel approach, Huang et al. targeted paclitaxel into the tumour cells by endocytosis of SSTR. Paclitaxel is known for its excellent antitumour activity by lacking cell specificity [211]. Huang et al. synthesized an OCT conjugated paclitaxel that internalized into the cytoplasm of tumour cells expressing SSTRs [211]. This conjugate induced apoptosis in MCF-7 cells by promoting tubule formation by retaining the biological properties of paclitaxel [211]. Furthermore, OCT was proposed to prevent breast cancer invasion and a possible treatment approach for metastatic breast cancer by modifying daunorubicin plus dihydroartemisinin liposomes in MDA-MB-435S cells and MDA-MB-435S xenografts nude mice model of breast cancer [212].

Watt and Kumar showed the co-expression of all SSTR subtypes with epidermal growth factor receptor subtypes (ErbBs) in human breast cancer cell lines [213]. ErbBs are variably expressed in different tumours; ErbB1 and ErbB2 subtypes are the most over expressed in tumours, whereas the ErbB3 and ErbB4 are the least expressed. In total, 40–50% of the breast cancer cases express ErbB1 and are linked with poor survival [214]. ErbB2 is the most over-expressed receptor in ~30% of breast cancer cases and is associated with poor prognosis [215,216]. Direct antitumour activity of SST and its analogs is also by it a virtue of either activating or suppressing various signal transduction pathways such as MAPK pathways, PI3K/Akt pathways, and phosphotyrosine phosphatases such as PTP1 and PTP2, which causes the induction of a cyclin-dependent kinase inhibitor (p27^Kip1^) leading to cell arrest [210,217,218]. Studies have also shown that SSTR targeted liposomes in combination with diacerein, resulting in the inhibition of interleukin-6, and this might serve for breast cancer therapy [219]. The variable colocalization of SSTRs with ErbBs in human breast cancer cells has opened up the possibility for an interaction between these receptor systems which could be exploited for a novel therapeutic approach for the treatment of breast cancer. SSTRs form functional heterodimers with other SSTRs and other GPCRs, and the novel receptor that is formed has enhanced pharmacological and signalling properties [220,221]. Zou et al. overexpressed SSTR1 and SSTR4 in MDA-MB-435S cells and reported that these two receptors might involve in cell cycle arrest through dimerization/activation [200]. Interestingly, the homo- or heterodimerization of ErbBs also initiates multiple downstream signalling pathways to promote tumorigenesis. In human breast cancer cells, SSTR1 and SSTR5 subtypes heterodimerize with epidermal growth factor receptor and modulate the downstream MAPK pathway in an agonist-dependent manner [198]. This study gives an extensive account of the downstream signaling pathways and unfolds a few of the many mechanisms involved for antiproliferative and proliferative effects [198]. Furthermore, in comparative studies, SST analogues, namely AN-162 with cytotoxic effect, exhibit a higher degree of tumour growth inhibition in comparison to doxorubicin alone in triple-negative breast cancer expressing SSTR subtypes [222]. In addition to the crucial role of radiolabeled SST analogs in breast cancer imaging, the SSTR antagonist has also been proposed as a potential candidate for breast cancer imaging [223,224]. Taken together, with all of these incidences, it is evident that SST and its receptors are of importance, and further advancements could be made to target breast cancer using specific agonist/antagonist combinations.

Amongst all SSTR subtypes, SSTR3 is the key receptor subtype that inhibits tumour cell proliferation via the induction of apoptosis. In contrast, SSTR2 induced cytotoxic and cytostatic effects and inhibited tumour cell proliferation. Previous studies from the author’s laboratory described that the overexpression of SSTR3 in MCF7 and MDA MB231 cells inhibits cell proliferation and blocks EGF-induced cell proliferation in comparison to cells with endogenous expression [53]. This study further elucidated the two different modes to exert an antiproliferative effect. The SSTR3 selective agonist induces apoptosis and increases the expression of PARP, whereas SSTR2 elicits a cytostatic effect and leads to increased expression of P21. Furthermore, SSTR2 and SSTR3 regulate intracellular signaling and apoptosis in receptor and cell-specific manner in breast cancer cells [50,51,53]. These observations anticipate a pronounced antiproliferative effect and indicate that the synthesized SSTR2 and SSTR3 chimeric molecules can be proven as potential drugs in breast cancer treatment.

## 21. Somatostatin, Somatostatin Receptors and Prostate Cancer

### 21.1. Prostate Cancer

One of the most prevalent cancers in the United States and Western countries is malignant prostate cancer [225]. Prostate cancer is the most frequently diagnosed type of complex tumour in man and a common malignant disease; it is the second leading cause of death in man globally. The older population is most affected by prostate cancer, and like many other cancers, it involves many risk factors. Poor and unpredictable prognosis approaches and a castration-resistant form pose prostate cancer as one of the major health issues. Previous studies recognized that basal cells are the primary sources of prostate cancer, whereas recent studies support the role of luminal cells in the origin of prostate cancer. In the past few years, many studies at the molecular and genetic levels have been conducted by different groups towards the understanding of prostate cancer biology. Efforts have been made in order to delineate the associated events related to disease initiation and progression. Although the mechanisms involved are not well understood, studies also support a possible link between newly formed nerve cells which migrate from brain to prostate in tumour progression and migration to other parts of the body [226,227]. Consistent with this study, Mauffrey et al. described how tumor growth and metastasis are promoted by neural progenitors; specifically, in human prostate adenocarcinoma, neural progenitors are involved in aggressiveness and recurrence [228]

### 21.2. Somatostatin and Prostate Cancer

SST plays an important role in the regulation of various physiological processes in different organs, including the prostate. The prostate glands neuroendocrine cells (NE cells) produce SST, which exerts its effects via paracrine regulation [202,229]. The most important function is the anti-neoplastic activity exerted by SST-14. In addition, SST and its analogues mediate its effect via increased apoptosis of cancer cells, decreased tumour cell growth and angiogenesis [69]. These regulatory effects are exerted either via receptors expressed on the tumour cells, i.e., direct effects mediated by its five receptor subtypes or indirectly on the normal cells by modulating with the rate of release of various growth factors, thus controlling cellular growth [230,231]. The expression of SSTRs has been shown on prostate tumour cells extensively; therefore, SST or its analogue critically regulates the growth of prostate tumours [232]. Previous studies using in vitro and in vivo experimental approaches have demonstrated SST analogues as an effective treatment in prostate cancer [233,234].

SST-14 analogues such as OCT have been successfully demonstrated as an antiproliferative agent in different tumour models, including the prostate and pancreas. These analogues are more stable and less toxic than regular adjuvant chemotherapy [11]. Peptide receptor radionuclide therapy (PRRT) using a high dosage of labelled OCT has been tried in prostate cancer. Nude mice bearing experimental human prostate carcinoma showed a higher uptake of Tc-99m-labelled SST analog [235]. The only limitation for PRRT in prostate cancer treatment is the lower expression of SSTR2 in prostate adenocarcinoma. So far, in clinical trials, only non-radiolabelled SST or its analogues are used to treat prostate cancer that also without any major success. [236]. For example, no tumour regression was observed in 24 patients with an advanced hormone-refractory tumour when treated with OCT; however, no toxic effect of the SST analogue was observed [237]. Moreover, blocking endogenous SST supports the antiproliferation effect of prostate cancer in an SHP-1-dependent manner [32].

## 22. Somatostatin Receptors and Prostate Cancer

The role of SSTRs in the regulation of prostate growth has been shown in many previous studies [238,239,240]. In normal prostate, neuroendocrine cells are involved in forming neuronal communication and regulating cell proliferation-independent androgen activation. Furthermore, studies support the presence of SSTR subtypes in these neuroendocrine cells. Ambrosini et al. reported that recognizing these receptors with labelled SST analogs has served as an effective tool in evaluating hormone-resistant prostate cancer [241]. The expression of SSTRs still remains controversial, with conflicting results reported by different groups; however, most of the studies reported the presence of SSTR2 expression in prostate adenomas and the absence of SSTR5 expression. The potential growth regulatory effect of SST is mediated via its five receptors, as they are present in both normal and tumoural prostate tissue [232]. Reverse-transcription PCR, real-time PCR and Northern blots were among the methods mainly employed in previous studies to detect the SSTRs from a tissue mass. Thus, the exact cellular location for the receptors cannot be determined until methods like immunohistochemistry or autoradiography are used for detecting the receptor expression in normal or malignant prostate tissue. Therefore, it remains critical to elucidate the expression and distribution of SSTR subtypes in the prostate to demonstrate the possible role of SST or its analogues.

Previously conducted studies demonstrated that SSTR5 was the least expressed or not expressed in normal or prostate tumours, whereas hSSTR2, -3 and -4 were variably distributed in the normal and malignant prostate tissue. Further, hSSTR1 expression was mainly confined in the tumour tissue, as well as in the neuroendocrine cells. The expression of SSTR1 has also been reported to be in the stromal compartments, in contrast to SSTR4 localization, which is mainly confined to the epithelial cells [242]. Intense staining for SSTR2 in the peritumoral vessels was reported, as SST might be involved there in regulating endothelial cells homeostasis and have an anti-inflammatory effect [232,242]. Selective expression of hSSTR1 in the cytoplasm of endothelial cells [243] was observed, and hSSTR2 expression was mainly confined in the stromal compartment [232]. Hansson et al. also reported that mRNA for SSTR2 was mainly localized in the stromal compartments [244]. High expression of SSTR2 in stromal cells was further proved by the efficient binding of octreotide to the stromal cells of the prostate cancer tissue [242]. Low mRNA expression of SSTR2 was observed in the prostate cancer cell lines PC-3, DU-154 and LNCAp [242]. Hennigs et al. further emphasized the significance of SSTR2 in prostate cancer. They reported that prostate cancer displaying loss of SSTR2 is a highly aggressive and proliferative and possible indicator of early metastatic and relapse [245]. SSTRs in smooth muscle cells possibly regulate the release of growth factors produced by stroma, as these growth factors, in a paracrine manner, act on the prostate gland to regulate its growth [242]. Immunohistochemical studies have reported the expression of all five receptors on the smooth muscle cells of prostatic stroma and showed that SSTR1 was the most expressed receptor, whereas SSTR5 was the least expressed receptor [243].

On the contrary to the observation of high levels of SSTR2 in prostate cancer tissue, Halmos et al. investigated 22 human prostate tissue samples, using RT-PCR, for mRNA expression and reported a detection rate of 86% for SSTR1 and 64% for SSTR5. There was a low expression of SSTR2, which was only detected in 14% of these cases [246]. SSTR1 has also been reported in prostate cancer as a diagnostic marker and as a therapeutic target [247]. An index of prostate cancer growth, the Gleason grade is negatively linked with the reduced expression of SSTR2, along with SHP-1, in prostate cancer [248]. Furthermore, studies have shown the reduced expression of SSTR2, along with SHP-2, in advanced prostate cancer cases, but the molecular mechanism associated with such loss in receptors protein and signalling pathways is not well known. Previous observations were made that the slice variant of SSTR5, namely sst5TMD4, is highly expressed at the level of protein and mRNA in prostate cancer and can govern some pathophysiological roles as a biomarker and/or therapeutic intervention in patients with worse prognosis [249]. Interestingly, some existing studies also support SST and its analogs’ role in combination with Docetaxel and metastatic castrate-resistant prostate cancer [250,251]. Mori Hiroshi et al. reported the determination of distribution and lesion, along with therapy effect, by using SSTR scintigraphy (SRS) in uncommon neuroendocrine-differentiated prostate cancer and proposed that SRS can provide information that is not normally not seen by using other conventional markers or image approaches [252].

## 23. Somatostatin Receptors and Pancreatic Cancer

### 23.1. Pancreatic Cancer

Pancreatic adenocarcinoma is considered as 5th leading cause of death in western countries, including the USA [253]. Mortality and the incident reporting rate remain the same for the Pancreas, which is one of the major concerns. In addition to multiple other factors involve in tumor growth and treatment failure, hypoxia is a critical determinant of tumor microenvironment. Furthermore, studies support that induction of certain genes which involve in tumor promoting factors, including invasion, angiogenesis, metastasis and metabolism, are linked to hypoxia in pancreatic adenomas [254]. The presence of excess tumor-associated hormones plays a crucial role in the diagnosis of a pituitary tumor. α-cells, β-cells, δ-cells and PP cells are the endoderms derived from four types of cells in the islets of Langerhans that constitute endocrine pancreas [5]. Glucagon, insulin and SST are major secretions from these cells. Cumulative evidence indicates that pancreatic adenocarcinoma is the 12th and 11th most common cancer with an incidence rate of 5.5/100 in man and 4.0 per 100,000 in women, respectively [255,256,257,258,259]. Selective gene mutation and changes in signaling pathways, as well as inactivation of factors linked to tumour suppression, are critical determinant of pancreatic tumour. Furthermore, besides age and genetic factors, several other factors, including tobacco smoking, heavy alcohol consumption, dietary factors and a pathological condition such as obesity and diabetes mellitus, have been associated with the progression of pancreatic adenocarcinoma [255,257,259]. Geographical distributional studies support the higher occurrence of pancreatic adenocarcinoma in developing countries when compared to the African and Asian populations. At present, there is no effective treatment for pancreatic adenocarcinoma.

### 23.2. Pancreatic Cancer and Somatostatin

SST is one of the most effective adjuvant therapies used to reduce the growth of the pancreatic tumour. Many studies have failed to report SSTRs’ expression in pancreatic adenocarcinoma, which is directly related to the loss in tumour growth regulation by using SST or its analogues. Pancreatic δ cells produce SST, which further exerts a regulatory effect in endocrine islets on the secretion of insulin, glucagon, SST and pancreatic polypeptide cells [260]. This effect is rendered via paracrine regulation and regulation of islets microcirculation. Studies suggest that intra-SST regulates islet hormone secretion [261,262,263,264]. SST-28 regulates insulin secretion more effectively than SST-14; on the other hand, SST-14 inhibits glucagon secretion more efficiently than SST-28 [265]. These regulatory processes occur at the level of gene transcription and post-translational modifications.

Furthermore, Ca^2+^, K^+^, glucocorticoid and insulin play a critical role in regulating SST secretion, as well as SST gene expression [266,267]. SST mRNA is induced by the low dose of glucocorticoids and inhibited by its high doses [268]. SST plays a critical role in the pancreatic development of the fetus and neonate. SST secretion starts after the gene expression of insulin and glucagon is over such that SST can regulate the growth of endocrine progenitor cells once the appropriate cell number has reached. Using an autofeedback mechanism, SST can also inhibit its own secretion; thus, it can regulate the growth of cells in adults [260].

### 23.3. Pancreatic Cancer and Somatostatin Receptors

SSTRs are widely distributed in different tissues with specific expression levels and colocalization with other proteins [269]. The information regarding the expression of SSTR subtypes in pancreatic cancer is limited, the data available are at the level of mRNA expression and SSTR expression at the level of protein is still awaited. mRNA expression of SSTR2 and SSTR5 has been shown in different pancreatic cancer cell lines including Panic-1, MIA PaCa-2 and Hs 766T. Tissue specimens of pancreatic cancer displayed decreased mRNA expression for SSTR2 and SSTR5 when compared with the normal tissue adjacent to the carcinoma. SSTR4 remained undetected, as week expressions for SSTR1 and SSTR3 were reported [270]. The poor stability of mRNA and lower transcriptional rates probably resulted in lower receptor expression, as reported by many biochemical and histochemical studies. Post-translational modification and alteration in the transport of these receptors to the surface might result in the loss of surface expression of these receptors. Further specific localization of mRNA was not demonstrated as from islets cells or nerve fibres, or from acinar cells. Ludvigsen et al. reported the quantitative analysis of the localization of SSTRs in mice and rat pancreas, specifically in islet cells and their colocalization with critical pancreatic hormones like insulin, SST and glucagon [271]. The occurrence of SSTR1, SSTR2 and SSTR5 is more readily reported than that of SSTR3 and SSTR4 [260]. In the human pancreas, all five SSTR subtypes are present with a mild expression of SSTR4 [272]. β-cells normally express SSTR1, and some also show the presence of SSTR5. Moreover, α-cells express SSTR2, and δ-cells express SSTR5 [272]. Thus, the distributional pattern of receptor expression indicates that SSTR5 plays an important role in inhibiting insulin secretion where as SSTR1 and SSTR2 regulate insulin and glucagon, respectively [273]. Similar studies reported in rodents also suggested the role of SSTR2 and SSTR5 in regulating glucagon and insulin secretions, respectively. SSTR1, -2 and -5 were expressed in the majority of insulin-positive β-cells in mice, but only 50% of rat pancreatic islet β-cells displayed their expression. Moreover, α-cells, which mainly constitute the periphery of the islet, displayed higher co-expression of SSTR2 and -5 with glucagon in both mice and rat, while γ-cells, which secret SST, were mainly found in the islets mantle area and displayed stronger co-expression of SST with SSTR3 and -4 in rat pancreas. SSTR1, -2 and -5 displayed similar colocalization in both mice and rats [271]. Specifically, high cellular expression of SSTR2 has been described in tissue from pancreatic inflammatory and tumoural tissues [274,275].

Previous studies have shown that SSTR2, in addition to the treatment of inflammatory diseases and arthritis, has shown some beneficial effects in acute pancreatitis, pancreatic adenocarcinomas and carcinoids [276,277,278]. Techniques like immunohistochemistry, autoradiography and scintigraphy were unsuccessful in detecting the high expression levels of SSTRs in pancreatic adenocarcinoma [279]. Thus, techniques utilizing SST or its analogues failed to regulate the tumour growth in vivo or in vitro. Fisher et al. demonstrated that the reintroduction of SSTRs into pancreatic cancer cell lines has a growth regulatory effect on these cells involving p16, p21 and p27 expression [279]. Decreased tumour growth was also reported in animal models where the SSTR2 gene was introduced artificially [279,280].

## 24. Somatostatin Receptors and Pituitary Tumour

### 24.1. Pituitary Tumour

Most of the pituitary tumours are benign in nature and involve different types of cells which specifically retain their original characteristic for secretions [281]. Pituitary tumours are classified on the basis of the kind of hormone that is produced and secreted by tumour cells. These hormones and types of the cells include corticotropic adenomas, which secrete adrenocorticotropic hormone (ACTH) (Cushing’s disease); somatotrophic adenomas secrete growth hormone (acromegaly); thyrotrophic adenomas secrete thyroid-stimulating hormone (TSH); gonadotropic adenomas secrete luteinizing hormone (LH) and follicle-stimulating hormone (FSH); and prolactinomas secrete prolactin (PRL). This is also observed that some of the neoplastic cells can secrete two hormones, including GH and prolactin, that serve to develop chimeric molecules for the treatment of pituitary adenomas associated with over-secretion of GH and PRL (discussed in detailed later).

### 24.2. Somatostatin Receptors and Pituitary Cancer

SST via binding to SSTR subtypes exerts an inhibitory role on pituitary growth. Inconsistent expression of SSTRs has been detected in different pituitary tumours, which resulted in a significant difference in response of SST/analogues efficiency to regulate tumour growth. SST was first reported as the inhibitor of growth hormone released by the anterior pituitary cells of rats as a component of the hypothalamic extract [282]. The mRNA of all SSTRs other than SSTR4 has been reported from the normal pituitary tissue. In normal fetal pituitary tissue, expressions of SSTR2 and SSTR5 were prominent, and they play a critical role in regulating GH secretion, as reported [283]. The presence of SSTR subtypes in pituitary tumours depends on the hormone secreted and the grade of the tumour. The suppression of tumour growth by SST and its analogs has also been reported independent of SSTR subtypes.

### 24.3. Pituitary Adenomas Secreting Adrenocorticotropin

Reubi et al. reported the absence of an SSTR2 binding site on the ACTH adenoma using a radiolabelled SSTR2 ligand [284,285]. Similar results regarding the absence of SSTR2 were reported when no increase in the level of radiolabelled OCT uptake was observed. Several reports using PCR have suggested the higher expression of SSTR5 in comparison to SSTR2. Using RT-PCR, studies have suggested the expression of SSTR1, -2 and -5 in most carcinomas with SSTR5 as a major receptor subtype at the mRNA level, similar to the protein expression levels. Thus, SSTR5, in terms of its expression at mRNA level or translated to protein, becomes a major target in ACTH adenomas. Colocalization of all 5 SSTRs has been shown with ACTH in normal rat pituitary corticotropes [286]. Furthermore, studies have suggested the ineffectiveness of SST on the basal or stimulated ACTH levels [287,288]. Studies have also demonstrated increased the ACTH level in SSTR5 ko mouse in comparison to wt. corticotropin-releasing hormone-induced ACTH and corticosterone secretion were inhibited by SOM230, which binds with higher affinity to SSTR2, -3 and -5 in comparison to OCT. Schonbrunn et al. reported the negative regulation of SSTR expression by glucocorticoids, especially SSTR2 [289]. Dexamethasone treatment in GH4C1 and AtT20 cells reduced SST binding by 20–40%, and prolonged treatment caused a significant decrease in SSTR1 and SSTR2 mRNA levels by 50 and 30%. Park reported an increase in SSTR5 mRNA expression as the SSTR1-4 level decreases. Treatment targeting SSTR1, -2, -3 and -5 using SOM230 (pasireotide) resulted in a more efficient regulation of ACTH secretion, providing the platform for Phase II clinical studies in Cushing’s disease.

### 24.4. TSH-Secreting Pituitary Adenomas

Like other types of pituitary adenomas, the expression of SSTRs in TSH-secreting adenomas was also observed, with specifically high expression of SSTR1 and SSTR2 [290,291]. Regulation of T4 and TSH levels has been observed in patients when treated with OCT [291]. A distinct response rate for octreotide has also been demonstrated [292,293]. High expression of SSTR2 has been exploited by using octreotide and Lanreotide to normalize the expression of T4 and TSH levels [294].

### 24.5. Prolactinomas

Prolactinomas, a most common and frequent type of pituitary adenomas, constitute almost 50% of all pituitary tumours. The over-secretion of prolactin from pituitary lactotroph is associated with prolactinomas. Surgical resection and radiotherapy were primary choice of treatment preference; however, the size and invasive nature of tumours and side effects limited the use of these approaches. Studies recognized dopamine as an inhibitor of prolactin release from pituitary cells lactotroph three decades ago. At present, dopamine agonists are the treatment of choice for prolactinomas, resulting in the suppression of PRL and cell proliferation [295,296]. Dopamine-negative regulation of prolactin is mediated by dopamine receptor type 2, which is well expressed at the cell surface of lactotroph and belongs to the GPCR family. However, dopamine agonists are not always successful in all patients, and in 10–20% of patients with prolactinomas, they failed to normalize PRL [296,297]. Using radiolabeled SST, the presence of multiple SSTRs has been demonstrated in prolactinomas. Studies involving RT-PCR have suggested higher levels of SSTR1 mRNA in comparison to SSTR2 and SSTR5. The expression of SSTR3 has also been reported from a few adenomas [298].

Furthermore, very low levels of SSTR4 expression were reported, and the expression of SSTR2 was reported to be lower than that of SSTR1 and -5 [298]. Studies have shown that SST or its analogues, such as SSTR2-specific OCT, do not significantly affect the regulation of TRH-stimulated PRL secretion [299,300]. In the presence of estradiol, SST or its analogues in normal lactotrophs were able to regulate PRL secretion. SST can inhibit PRL release in men post-treatment with estrogen [301].

### 24.6. GH secreting Pituitary Adenomas: Acromegaly

Acromegaly, a chronic metabolic disease, is characterized by the over-secretion of GH by pituitary adenomas, resulting in the increased release of IGF1 from the liver. Large numbers of acromegaly patients are sporadic GH secreting adenomas that emerged either from somatotrophs or from cells producing GH and prolactin. Despite the significant progress in diagnosis and prognosis, prolonged exposure to hypersecretion of GH and IGF1 resulted in multiple comorbidities, as well as somatic disfigurement and systemic manifestation [302]. The signalling pathways linked to acromegaly are also involved in mutation in guanine nucleotide-binding protein G(s) (GNAS1), the mutation that confers preferential responsiveness to GH inhibitory peptide SST [302,303]. The second signaling molecule associated with GH over-secretion is the signal transducer and activator of transcription 2 (STAT3). In addition, the presence of germline mutation in the gene encoding aryl hydrocarbon receptor-interacting protein (AIP) in young patients with acromegaly and patients with pediatric syndrome X-linked acrogigantism added some new information in understanding of acromegaly [302,304,305,306]. Furthermore, GH over-secretion might be associated with the abnormal response of GH-producing cells in the pituitary to a glucose-dependent insulinotropic polypeptide (GIP) or overexpression of GIP receptor (GIPR) as proposed earlier [297,307]. Furthermore, studies also support cell cycle disruption, which is often seen in GH-secreting pituitary tumours [297].

### 24.7. Somatostatin Receptors and Acromegaly

Increased IGF1 and GH serum levels are the diagnostic tools for acromegaly. Trans-sphenoidal surgery is the first effective treatment choice to achieve suppression in GH levels in acromegaly patients. However, on the basis of differential binding of SST and its analogues, the presence of different SSTRs in GH adenomas has been suggested [285,308]. The presence of mRNA for all the SSTRs except SSTR4 has been reported in GH adenomas. Studies focusing on the quantitative expression of SSTR mRNA have shown the presence of SSTR2 and -5 in most of the GH adenomas with higher expression levels of SSTR5 [309]. Accumulated evidence from previous studies indicates that SSTR2 is highly expressed in and followed by SSTR5 and SSTR2 and SSTR3. At the same time, expression levels of SSTR4 have not been detected. Furthermore, SSTR2 expression levels are directly associated with GH suppression in patients treated with SST analogs. In most patients with first-generation acromegaly, SSTR ligands (SRLs) are the first line of therapeutic drugs in addition to DA agonists and second-generation SRLs [310]. OCT as primary Sandostatin-LAR or SR-Lanreotide as secondary therapy has been used widely in GH adenomas [311,312]. Regulation of GH to its normal levels is normally achieved with SST treatment or its analogues and the significant tumour growth retardation [311]. The sensitivity of patients to this treatment is normally variable because of the differences in the receptor expression on these adenomas. SSTR2- or SSTR5-specific SST analogues were highly effective in GH secreting adenomas, with an additive effect observed when SST analogues specific to both SSTR2 and SSTR5 were used [313]. Pasireotide is approved clinically for the treatment of acromegaly displaying binding to multiple SSTR subtypes in order of SSTR5 > SSTR2 > SSTR3 > SSTR1. Studies also support different functional properties of pasireotide from SST and first-generation SST ligands. Early treatment is essential because acromegaly left untreated resulted in severe complications [314,315]. Furthermore, patients with acromegaly also displayed impaired cognitive function, as well as psychosocial abnormalities [314,316,317]. Taken together, all the information available regarding the use of SST analogs with variable efficacy in the treatment of pituitary adenomas shows that they play a significant role in tumour shrinkage and suppression of tumour size in a large number of patients.

### 24.8. Gonadotrophs: Non-Functioning Pituitary Adenomas

Non-functioning pituitary adenomas (NFPAs) are a heterogenous group of pituitary adenomas that are not hormonally active and constitute 15–30% of pituitary adenomas. In comparison to functioning pituitary adenomas these pituitary adenomas are benign and lack associated clinical symptoms. A majority of NFPAs are believed to emerge from gonadotroph cells. Interestingly, in contrast to the secretion of gonadotropins, these pituitary adenomas are devoid of in vivo hormonal secretion despite the possibility of presence of hormones including GH, PRL TSH or ACTH determined immunohistochemically [318]. Studies have shown the aggressive nature of NFPA, albeit in limited cases with invasion and recurrence post-surgery [319]. It is believed that these adenomas are diagnosed late as invasive macroadenomas and not associated with clinical syndromes often seen with the overproduction of pituitary hormones including amenorrhea–galactorrhea, acromegalic features, hypercorticism or hyperthyroidism. The majority of these tumors exhibited symptoms which are associated with mass effects like visual defect, hypogonadism and headache [320].

### 24.9. Somatostatin Receptor Expression in Non-Functioning Pituitary Adenomas

Studies conducted by different groups have shown the presence of SSTRs in membrane preparations of gonadotroph adenomas. It has also been demonstrated that the binding sites for radioactive OCT clearly indicate the presence of SSTR2 [321,322]. Previous studies describing the presence of SSTR subtypes in NFPA support the expression of SSTR subtypes at the level of mRNA and protein [319,323,324,325]. Gonzales et al. showed high expression of SSTR3 followed by SSTR2, -5 and -1 by qPCR and further emphasized increased expression of SSTR3, -2 and -5 at protein level by immunohistochemistry in NFPA [326]. Meanwhile, Zawada and Kunert-radek reported the presence of all SSTR subtypes with a common expression of SSTR5, -2 and -1 with strong expression of SSTR1 and SSTR3 [327]. Taboada et al. also showed high mRNA expression for SSTR3 in NFPA cases [328]. Previous studies have also reported comparative distribution of D2R and SSTR subtypes mRNA in NFPA with high expression of D2R than SSTR subtypes [329,330]. One of the same studies also revealed that D2R expression is positively correlated to SSTR2 and SSTR3 and negatively to SSTR1 and SSTR5 [329]. The expression of SSTR1-5 and DR subtypes at the level of mRNA and proteins has also been described in NFPA with high expression SSTR3 and D2R subtype [319]. In general, it is believed that SSTR subtypes are well expressed in NFPA, with the exception of SSTR4. Taken together, growing evidence supports the presence of DR and SSTR subtypes at the cell surface exhibiting distinct distributional pattern and intensity supporting possible role of dopamine agonist and SST analogues in treatment of NFPA.

### 24.10. Non-Functioning Pituitary Adenomas and Treatment Preferences

NFPA are challenging to treat, and transsphenoidal surgery is the choice of first-line treatment for symptomatic patients. Moreover, the invasive nature of the tumor, the impossibility of completely resecting the tumor and the tumor regrowth that takes place in a large number of cases pose an urgent unmet need to treat such adenomas [320,331]. In a quest for treatment of NFPA, SST analogues, dopamine agonist and analogues for GnRH have been proposed with controversial results. In contrast to lack of SST analogues response, multiple studies support the beneficial role of the SST agonist in a receptor-specific manner in NFPA [332]. Gonadotropins or their α-subunit secretions have been shown to be inhibited by the SST and OCT [322,333]. It has been demonstrated that SST, OCT or lanreotide in vitro inhibits the proliferative gonadotrophs [334]. Although previous studies have suggested that SSTRs other than SSTR2 are involved in this type of adenomas, studies involving other SST analogues like SOM230 should be further evaluated, as high expression levels of SSTR3 in these adenomas are of great relevance [335]. Florio et al. showed significant inhibition of cell proliferation with the use of SST and lanreotide in NFPA, thus opposing Fusco et al.’s in vivo studies using OCT LAR [334,336]. In conclusion, SSTR subtypes’ expression in NFPA is associated with reduced cell viability in a receptor-specific manner [337]. Furthermore, an interesting observation emerged from Zatelli et al.’s studies, showing the inhibition of VEGF-induced cell proliferation upon treatment with pasireotide [338]. Unlike the very effective and established treatment of acromegaly, SST analogues are not as effective in the treatment of NFPA. Zawada et al. reported that when patients were treated with SST analogues prior to surgery, it resulted in a better outcome [339]. Furthermore, the use of post-surgery radiotherapy was proposed to minimize the risk of tumour recurrence or regrowth [320]. Although all SSTR subtypes are expressed in NFPA and efficacy is linked to the expression of receptor subtypes, the question of debate is whether it is SSTR1 or SSTR3 that is highly efficacious. Recently, using a NFPA rat model, SSTR3’s new full agonist, namely ITF2984, was shown to exhibit antitumor activity [340]. Like SSTR subtypes, the efficacy of D2R was also determined with the expression level of D2R. However, studies support that D2R agonists are only effective once used after surgery.

In prolactinomas, dopamine agonist suppressed hormone release and tumour shrinkage, but such an effect in clinically NFPA is disputed despite the comparable distribution of mRNA for D2R in NFPA [330]. The dopamine agonist mediated a direct and indirect antitumor effect, with the antiangiogenic effect supporting therapeutic intervention in the treatment of NFPA [341]. Furthermore, the D2R expression label is a determinant factor for such an effect. Although such treatment options come with only a minimal benefit in improving tumor size, they suggest that therapeutic intervention in NFPA treatment in combination with dopamine agonist and SST analogues might be more effective than a single treatment, as reported earlier [330,342]. This notion is further supported by previous studies describing the interaction between SSTR5 and -2 with D2R in a heterodimeric complex formation [343].

The expression of D2R and SSTR subtypes in NFPA is an indication for the role of DA agonist and SST analogues in combination for the treatment of NFPA. Past studies, in addition to the inhibition of gonadotropins secretion in in vivo and in vitro, have also shown tumor shrinkage with the use of DA agonist and SST analogues either used alone or in combination [344,345]. In contrast, there are no successful treatments and multiple studies linked to the expression level of receptor subtypes, as well as the intensity of receptor homo- and heterodimerization [328,346]. Since the number of NFPAs displaying high expressions of SSTRs, as well as colocalization between SSTR and DR subtypes, are very low, researchers have argued for the lack of an additional effect on inhibition cell proliferation in the presence of chimeric molecules than DA agonist alone [345,347]. Altogether, SST analogues and DA agonists govern significant preference in treatment of NFPA in case of limited access to surgery and radiotherapy failure.

### 24.11. Recent Developments of Chimeric Molecules in Treatment of Pituitary Tumour

In patients with acromegaly, the SSTR2-specific agonists OCT and lanreotide reduce the GH secretion. [309,348]. Previous studies support the presence of other SSTR subtypes in pituitary tumours in addition to SSTR2. Importantly, the simultaneous expression of SSTR2 and SSTR5 might enhance signaling pathways mediated by SSTR2 through receptor heterodimerization [349]. The role of SSTR2 and SSTR5 heterodimerizations with enhanced response in acromegaly is further supported with observation describing the weak response of SST ligand in acromegaly with low SSTR2/SSTR5 ratio, as well as in the presence of the Sst5TMD4-truncated isoform of SSTR5 [350,351,352]. In addition to SSTR agonists, in patients with relatively low expression levels of GH and IGF1, dopamine agonists have also been used as a treatment preference. Moreover, in combination with SSTR-specific agonist, dopamine agonist has also been used in patients showing resistance to SRLs. Furthermore, in support, studies have demonstrated the expression for dopamine receptor 2 (D2R) in these tumours and correlated with the tumour suppression using D2R agonists with variable D2R expression levels [348]. SSTR and D2R agonists, when given together, exert additive effects and have been used in patients with acromegaly [353]. SSTR and D2R agonists resulted in developing chimeric molecules with therapeutic potential to target SSTR5 and dopamine receptors (D2R). Several studies in the past have reported that SSTR2 and D2R heterodimerize and form a signalling unit with novel properties. The formation of heterodimers between SSTR2/D2R and SSTR5/D2R has been shown with changes in pharmacological properties than with the native receptors [220,343]. Previous studies have demonstrated that chimeric molecules which can activate heterodimers of SSTRs and dopamine receptors called DOPASTATINS appeared to be more efficacious [354]. Furthermore, studies also support that the SSTR2/D2R chimeric molecule was more efficient in regulating GH secretions than using SST and D2R drugs separately [347,354,355]. Importantly, incorporating SSTR5 affinity to these dopastatins has emerged as an effective therapeutic approach and an option for treating acromegalic patients [354]. Although significant progress has been made for the treatment of acromegaly in the last few years with the surgery as a mainstream treatment, treatment preference for acromegaly is still a concern, specifically in the case of aggressive and resistant pituitary tumours.

## 25. Somatostatin and Lung Cancer

### 25.1. Lung Cancer

Lung cancer is the major cause of cancer-related deaths globally, and it accounts for more than a million deaths annually [356]. Lung carcinogenesis is a multistage process involving defects in multiple genes and signalling pathways [357]. There is no effective treatment for lung cancer, probably due to the lack of adequate diagnosis at the early stage of the disease. The available treatment options, including surgery, radiation and platinum-based combination chemotherapy, have achieved limited success, and the disease is associated with a poor prognosis [358]. However, studies are very limited, and selected populations have shown promising results by using small-molecule approaches. Lung cancer has been classified into two different types: small cell lung cancer (SCLC) and non-small cell lung cancer (NSCLC). NSCLC includes squamous cell carcinoma, adenocarcinoma and large cell carcinoma and diagnosed post occurrence of metastases condition which is beyond successful therapy. Among all types of Lung cancer, NSCLC accounts for approximately 80–85% of cases [359]. On the other hand, SCLC is a high-grade, poorly differentiated, lethal and aggressive lung cancer that constitutes 15% of diagnosed lung cancer and accounts for 25% of lung cancer-associated death. SCLC is the type of lung cancer which not only associated with poor survival but also metastasizes at an early stage [360]. In addition to SCLC and NSCLC, typical and atypical carcinoids have also been reported in pulmonary neuroendocrine tumour [361].

### 25.2. Somatostatin Receptors Expression and Role in Lung Cancer

The expression of SSTR subtypes has been reported in human SCLC cells, suggesting a possible role of SST in modulating tumour growth [362]. SSTRs are widely distributed in humans, and SSTR4 levels are in particular found to be higher in the lungs [5,203]. In addition, a high density of SSTR2 has been found in bronchial tumours and SCLCs [363]. As shown earlier, all five receptor subtypes are present in lung cancer and exert a significant role; however, SSTR2 is the prominent receptor subtype with critical pathological significance in lung cancer. Furthermore, in comparison to low-grade tumours, relatively high expression of SSTR2 and SSTR3 has been reported in low intermediated groups of SCLC [364]. The presence of SSTR subtypes in lung cancer leads to the use of SST analogs in the treatment of SCLC and has been described earlier [365]. It was found that BIM-23014C, an SST analogue, suppressed tumour progression by 59% vs. the control in the human SCLC cell line, NCI-H69. Another study found that long-term treatment of NCI-H69 with a long-acting SST analogue MK-678 decreased the tumour area, DNA and RNA content compared to the control [366]. SCLC also leads to the generation of some endogenous peptides like Bombesin or Gastrin-Releasing Peptide (GRP). GRP behaves like autocrine growth factors and activates the tumour’s growth [367,368,369]. SST analogues have been found to reduce the production of Bombesin and, hence, might play an important role in SCLC treatment [238]. When H-69 SCLC cell line xenografts expressing mRNA for SSTR2 were introduced in nude mice and subsequently treated with an SST analogue AN-238, tumour growth was inhibited [370]. Amongst different SSTR subtypes, SSTR2 has drawn significant attention in SCLC, and multiple studies have described significant contributions. However, the results with the use of SSTR2 agonist in clinical trials are not encouraging. A cohort of 96 patients with SCLC described that 48% of patients expressed SSTR2 and established an important link between SSTR2 expression levels and clinical output [360]. Furthermore, this study emphasized that the downregulation of SSTR2 resulted in enhanced apoptosis and suppressed tumour growth that was linked to the negative regulation of AMP-activated protein kinase-α (AMPK-α) and increased oxidative metabolism and suggests that SSTR2 is a poor prognostic biomarker in SCLC [360]. In SCLC cells, increased apoptosis has been described with the use of SST before and after chemotherapeutics drugs [371]. The use of SSTR2-induced cytotoxic effect and Notch signalling regulator valproic acid has been proposed in combination as a potential therapeutic intervention in SCLC by suppressing tumour growth and increasing SSTR2 expression. Also, studies indicate that SSTR2-mediated signalling is involved in tumour growth and lung cancer survival, which was further linked to highly active metabolism and loss of apoptotic signalling due to lack of P38 and Rb in SCLC [360]. Therefore, with these results, it was believed that the lack of therapeutic implication of SSTR2 agonist might be due to lack of P38 and Rb and the use of SSTR2 antagonist might serve as potential target.

SSTR subtypes are also expressed on peritumoral vasculature in NSCLC [370,372,373,374]. AN-238 administration in NSCLC cell lines H-157 and H-838 markedly reduced the tumour growth [370,372,375]. OCT, a SST analogue, has a higher affinity for SSTR2 and SSTR5, as demonstrated in human NSCLC cell line A549 [376]. Paclitaxel is a well-known anticancer agent that acts by promoting tubulin assembly into microtubules [377], as well as disrupting the cell cycle in the G_2_ and M phase [378]. Octreotide–paclitaxel conjugate demonstrated cytotoxic activity in an SSTR-positive A549 cell line but not so in SSTR-negative fibroblasts [379], which further confirms the potential role of SSTRs in cancer therapeutics. A previous study [376] confirmed that paclitaxel-OCT conjugate was more efficient in reducing the tumour growth than paclitaxel or OCT alone and in a manner suggestive of apoptosis. DNA fragmentation assays were performed to detect paclitaxel–octreotide conjugate-induced apoptosis in the human NSCLC cell line A549.

A recent study [380] revealed that SSTRs and dopamine receptors (DRs) are variably expressed in certain bronchopulmonary neuroendocrine tumours (BP-NETs). Chimeric compounds selective for both SSTRs and DRs were effective in modulating the proliferation and signaling in BP-NET cell lines NCI-H720 and NCI-H727. A similar study was performed by Ferone et al. in an NSCLC cell line Calu-6 co-expressing SSTRs and DRs, using SST, SST agonists, DR agonists and chimeras for SSTRs and DRs [381]. It was concluded that SSTR/DR chimeras were effective than the individual SSTR and DR agonists in inhibiting tumour cell growth, suggesting the involvement of SSTR/DR heterodimerization [381]. Furthermore, Claudia Pivonello et al. described the use of SSTR2 agonist octreotide and D2R agonist cabergoline in combination with mTOR inhibitor in lung cancer cells NCI-H727 [382]. This is in agreement with an earlier study by Rocheville et al. (2000) confirming the formation of heterodimers between SSTR2 and DR5 with enhanced functional activity [220]. The same approach is also currently being used in the treatment of pituitary tumours, acromegaly. Moreover, studies by using SSTR2 for delivery of nucleotide to NSCLC indicate that due to low expression, SSTR2 targeting DOTATATE is not efficient to delivery [383].

SST analogues also play a useful role as diagnostic tools in detecting various neuroendocrine lung tumours. The technique immunoscintigraphy uses SST analogues like OCT, Lanreotide or Pantetreotide and is based on the fact that SSTRs are overexpressed in carcinoids [384,385,386]. The expression level of SSTR subtypes might play a determinant role in drugs uptake and therapeutic delivery. Because of subtype-selective expression of SSTRs in lung cancers, receptor-specific targeted chemotherapy may provide more effective results and minimal dose-related adverse effects as seen in conventional chemotherapy.

## 26. Somatostatin and Liver Cancer

### 26.1. Liver Cancer

Hepatocellular carcinoma (HCC) or primary liver cancer with high mortality is the third leading cause of cancer-related death worldwide [387]. The annual projected deaths due to HCC are more than 500,000 [388]. Reynaert and Colle, summarized WHO 2018 statistics analysis and reported that liver cancer is the sixth frequent type of cancer with the 4th leading cause of cancer-related death worldwide and also indicated an increasing trend of incidence and death rat [389]. Many other primary cancers metastasize into the liver, the most notable being the colorectal cancer [390]. At the early stage of tumour surgical eradication of the tumour is the treatment of choice in 30% of the HCC cases. However, a majority of patients (70%) diagnosed with non-resectable HCC are associated with poor prognosis and with limited therapeutic options that poses challenges to health care personals [391,392]. In such patients, Cirrhosis is the possible cause of complication in treatment option. Pre-existing chronic liver disorders, as well as decreased hepatic reserves, greatly reduce the chemotherapy options. Monotherapy with a single chemotherapeutic agent has not been of much beneficial effects [393]. On the other hand, combination chemotherapy using cisplatin, doxorubicin, 5-fluorouracil or interferon-alpha may improve the response [394], but the risk of multidrug adverse toxic effects due to associated hepatic disorders is very high [395]. The diagnosis of HCC is usually based on a combination of clinical and laboratory features together with radiographic and histopathologic findings. HCC may arise from a chronic liver cell injury, triggering an inflammatory response followed by hepatocyte regeneration, liver remodelling, fibrosis and liver cirrhosis [396]. In addition to liver cirrhosis, the major risk factors of HCC are hepatitis B and C, which are believed to be involved in changing the genetic make-up and finally leading to cancer [387]. Because of the greater association of liver cirrhosis with HCC, surgical resection is impractical in more than 70% of HCC cases [397]. Previous studies have failed to achieve promising benefits using different chemotherapy combinations aimed to reduce associated toxicity and increase therapeutic efficacy [393].

### 26.2. Somatostatin Analogues: Choice of Treatment for Hepatocellular Carcinoma

SSTRs are variably expressed in HCC and may form a basis for designing targeted drug therapy. Growing evidence supports the expression of SSTR subtypes in liver cancer and their correlation with recurrence, prognosis and survival in HCC [389]. Several previous studies support the presence of SSTR subtypes at different densities in HCC [398,399,400,401,402]. Previous studies also support interesting correlation between and the expression of SSTR subtypes in HCC and markers of poor prognosis, as well as some key indicators of poor survival with SSTR membrane expression [389]. In HCC, the expression of SSTR is in an order of SSTR5 > SSTR3 > SSTR1 and SSTR2 whereas HCC is devoid of SSTR4 expression [403]. Hepatic oval cells (HOCs) are known to be involved in liver regeneration and possibly hepatic carcinogenesis [404]. An upregulation in the levels of SSTR1, -2, -4 and -5 were found in HOCs vs. hepatocytes and normal liver [405]. SSTR4 is the only subtype that is absent in hepatocytes and normal liver but well expressed in HOCs, suggesting a definitive role in HOCs migration as a chemoattractant [405]. Several previous studies have described the role of SST and its analogues in hepatocellular carcinoma [57,389,406,407]. OCT is well tolerated in HCC patients with pre-existing liver disorders and commonly used to treat patients suffering from variceal bleeding owing to its portal pressure-reducing ability [395]. In a clinical trial conducted on HCC patients, OCT significantly improved patient survival [408]. OCT blocks cell proliferation in human hepatocyte-derived cancer cell line HepG2 via SSTR2, -3 and -5 and in a PTP-dependent pathway [409]. SSTR2 is predominantly expressed intracellularly, while SSTR3 and SSTR5 are well expressed on the membrane [409]. In liver cancer cells, the inhibitory or blocking effect of OCT has been shown on invasion and metastasis through the upregulation of phosphatidylethanolamine-binding protein 1; matrix metalloproteinases; and their tissue inhibitors, metalloproteinase-2 and E-cadherin and suppression of matrix metalloproteinase-2 [410]. OCT might be associated with the reduced incidence of recurrence and metastasis post-surgery of liver cancer [410]. Altogether, SSTR2 and SSTR5 are most effective and prominent receptor of choice for treatment of liver cancer. The role of other receptors and their ligands working in tandem with SSTRs cannot be ruled out. Opioids normally act via opioid receptors, which are GPCR’s. Opioid analogue ethylketocyclazocine (EKC) displays antiproliferative and apoptotic properties in HepG2 cells by acting on SSTRs through the similar PTP signaling pathway as seen with OCT [411].

It is also worth noting that cortistatin, an inhibitory neuropeptide, is secreted by hepatocellular carcinoma cells and plays a role in the internalization of SSTRs and may thus modify SSTR functions. SST analogue AN-238 is a cytotoxic conjugate comprising 2-pyrrolidine-doxorubicin (AN-201) and a SST/OCT carrier (RC-121), is very effective in inducing apoptosis in liver cancer cells [412]. The potency of AN-238 is about 500–1000-fold more than doxorubicin alone, and there was no evidence of cross-resistance with doxorubicin [412]. Importantly, in less aggressive liver tumours, the association of SSTR5 has been shown for better treatment response [413]. It is interesting to note that pasireotide has also been used as the second line of treatment in patients with advanced stages of HCC [414]. Although results are controversial, accumulated evidence from multiple clinical trials since 2002 has shown the beneficial effect of SST and its analogues, including suppression of tumour, stability, overall survival and improvement in general quality of life.

Previous studies support the improved survival and quality of life in addition to the suppression of tumours [406,408]. Further studies need to be performed to evaluate the exact expression of SSTRs in various liver tumours and individualize the targeted therapy according to the receptor subtype expression profile. Receptors homo- and heterodimerization studies of SSTR’s within their own family and among other GPCR’s in liver cancer cell lines could also lead to insights into new therapeutic approaches.

## 27. Somatostatin Receptors and Thyroid Cancer

### 27.1. Thyroid Tumour

The thyroid gland is one of the major endocrine tissues that secrete and synthesize two hormones, T3 and T4 and play a crucial role in regulating hormonal maintenance and homeostasis. Impaired hormonal balance in the thyroid disrupts metabolic activities and leads to neurological and other different types of cancer. The thyroid is the most prone endocrine tissue to develop a tumour, and follicular epithelial cells are the site of tumour origin for most thyroid cancers. The incident rate of thyroid tumour, which is the most common endocrine malignancy, are gradually increasing worldwide. In addition to the complicated signaling pathways, a close relation between thyroid tumour progression and altered levels of growth factors including IGF, VEGF, FGF and associated receptors has been well characterized. Previous studies have also described antiapoptotic and activation of DNA synthesis upon activation of the GH receptor, specifically IGF. These observations support that overexpression of IGF might be associated with poor prognosis of some of the thyroid tumours. Based on previous histopathological evaluation and origin, thyroid tumours are divided into four different types, i.e., papillary thyroid cancer (PTC), follicular thyroid tumour (FTC), medullar thyroid tumour (MTC) and anaplastic thyroid tumours (ATC). In the thyroid gland, PTC, FTC and ATC often develop from endodermally derived follicular cells.

### 27.2. Papillary Thyroid Cancer

PTC is the most common thyroid tumour type and accounts for 80% of thyroid malignancy. PTC is easy to treat; however, it exhibits a variable size, including the formation of papillae and changes in nuclear morphology. PTC often has a tendency to spread to other regions via the lymphatic in the thyroid itself and is also known as a multifocal disease. Half of the PTC cases at the early stage of diagnosis have shown nodal metastasis. Furthermore, studies have also reported variants which might pose some limitation in prognosis.

### 27.3. Follicular Thyroid Cancer

FTC, based on tumour invasiveness, has been categorized in two forms. The major form of FTC includes the type of tumour that exhibits a lower rate of invasiveness, whereas in contrast, the second form is widely invasive. The occurrence of FTC amongst all thyroid cancer is further divided based on iodine deficiency. Studies revealed that 5–10% of FTCs are non-iodine deficiency areas, whereas larger cases which composed 30–40% belong to iodine-deficient regions [415,416]. A further morphological analysis revealed the presence of encapsulated (less invasive type) and non-capsulated FTCs, which are mostly invasive. FTCs devoid of encapsulation frequently invade regions in the vicinity of the thyroid, as well as blood vessels [415,416]. Although controversies exist, some studies support that FTC is like Hürthle cell tumour; in contrast, others claim that it has a distinct nature [416,417,418,419,420]. Age is the major determinant of life expectancy with FTC.

### 27.4. Medullary Thyroid Cancer

In the thyroid, calcitonin-secreting parafollicular C cells are the site of origin for MTC, and enhanced levels of calcitonin have been used as a true marker of MCT and index of the tumour. Contrary to differentiated thyroid tumour, MTC constitutes 5–10% of all thyroid tumours, which exhibit a slow growth rate but are considered the second most aggressive thyroid tumour [421]. Previous studies have also demonstrated that 20% of MCTs are hereditary, whereas 80% of tumours have a sporadic in nature. The occurrence of lymph node metastasis at an early stage of tumour growth is believed to be the most prominent and leading cause of negative prognostic characteristics of MTC. The large numbers of MTC are sporadic, and mutation in Ret proto-Oncogene for MTC has also been reported [421,422].

### 27.5. Anaplastic Thyroid Cancer

In all thyroid cancers, ATC accounts for 3–5% and is believed to be the most aggressive thyroid malignancy. ATC is frequently seen in the elderly population in the mean age group of 60–65 years old [415]. There is very limited therapy to treat ATC. With poor prognosis and traditional therapy devoid of any effectiveness against ATC, which is associated with the local aggressive nature of tumour and high rate nodal and distant metastases, including lung bone and adrenal [423,424,425,426,427,428]. ATC is a very rare type of thyroid tumour; however, it often originated in people who had a history of pre-existing PTC and FTC. Furthermore, ATC represents the most complex and heterogeneous morphological changes with a high rate of mitotic activities with widespread distribution in the thyroid itself, as well as some other body parts [415]. Previous studies also indicate that ATC often emerges from a pre-existing tumour that has been removed over an extended period [415].

### 27.6. Somatostatin Receptors Distribution in the Thyroid Gland

The presence and distribution of SST and SSTR subtypes in control and thyroid gland with tumour have been most controversial. SST-positive cells are frequently seen in medullary thyroid cancer in contrast to no expression in normal thyroid. Like any other endocrine tissue, SSTR subtypes are well expressed in the thyroid in a receptor-specific manner and display site-specific distributional patterns [429,430,431,432,433]. In addition, species-specific distribution has also been reported for SSTR subtypes in the thyroid. Furthermore, whether the thyroid tumour is medullar or non-medullar, the presence of SSTR subtypes might serve crucial forecaster of thyroid tumour. SSTR subtypes are the potential target of SST analogs to elicit multiple functions, including antiproliferative effect also exerting inhibitory role in angiogenesis and inhibition of several other downstream signaling pathways associated with tumour growth. In the thyroid, SSTR2 has drawn great attention because it binds strongly to OCT and Lanreotide, as well as pasireotide, which has frequently been used in the treatment and detection of tumours [434,435,436]. We recently described the comparative subcellular expression of all five SSTR subtypes in the rat thyroid gland [437].

### 27.7. Role of Somatostatin and Somatostatin Receptors in Thyroid Tumour

Thyroid tumours represent the most complicated morphological, biochemical and molecular abnormalities, which render tumours difficult to treat. The successful and improved prognosis of thyroid tumour relies on early diagnosis, specifically for MTC. Furthermore, integrated signaling pathways in the presence of several receptors and transporter protein and a wide range of functional interaction between different types of signal transduction pathways often account for the failure of treatment [438,439,440]. Differentiated, MTC and ATC differ in their origin, as well as prognosis. PTC and FTC are easy to treat and curable thyroid tumour and can be identified using radioactive iodine, and thyroidectomy is the best clinical intervention in the treatment/removal of these tumours with a good extendable survival rate.

In contrast, MTC, which is devoid of natrium iodide gene and failed to take up radioactive iodine that poses these tumours difficult to treat and resulted in poor prognosis. Furthermore, frequent unsuccessful rates of different treatment approaches, including chemotherapy, surgery, radiation and inhibitors for RET and MEKs, often resulted in poor clinical outcome. Furthermore, with the denial of chemotherapeutic drugs, inhibitors of receptor tyrosine kinases are used for MTC treatment [421,441]. In comparison to PTC/FTC and MTC, thyroid tumours and ATC are aggressive in nature and exhibit faster growth rate and metastasis to other regions and shorter survival [442,443,444]. Patients with ATC malignancy often resulted from poor prognosis and failure in treatment. Surgical approaches are in use to resect these tumours and to further remove residuals, and metastasis radiotherapy and chemotherapy are used to prolong survival and to block the growth of the tumour. Importantly, due to poorly understood resistance mechanisms, common chemotherapeutic approaches using doxorubicin, cisplatin, paclitaxel and 5-fluorouracil are not successful. Previous studies further emphasized that pumping out the therapeutic drug by resistance-associated proteins possibly accounts for resistance to chemotherapeutic drugs.

From the discussion above, it is obvious that multiple downstream signaling pathways and the complex nature of thyroid tumours often result in treatment failure and resistance to common chemotherapeutic approaches. While therapeutic approaches to treat thyroid tumours are limited, options are often associated with a substantial degree of toxicity and failure in treatment. The use of peptide receptor radiolabeled compound is well established, and SST is one that has been used frequently for multiple types of tumours. Including medullary and non-medullary thyroid tumours. SST, a growth hormone inhibitory peptide, emerged as an efficient peptide with an antiproliferative effect in most peripheral tissues and in vitro cultured cells, specifically in tumours of different origins. These observations further extended to examine the antiproliferative effect of the SSTR subtype in normal and tumour thyroid tissue. Several previous studies have shown the use of radiolabeled SST analogues clinically for the treatment of MTC and DTC [429,445,446]. One of the promising therapeutic interventions in thyroid tumour is SSTR recognition using internal radiation therapy. SST works via binding to five different receptors and exerts antiproliferative effect and induced apoptosis that account for potential therapeutic approaches in cancer treatment, including thyroid tumours. Moreover, the presence of SSTR subtype 1 has also been proposed to have a better therapeutic response in MTC [447]; however, SSTR1 is well expressed in most thyroid cancers in comparison to SSTR2. However, SSTR2 expression in the thyroid holds some clinical benefits. Lisa H. de Vries reported longer survival of stage IV MTC patients with the SSTR2 expression [448]. In addition, studies have also demonstrated the association of lymph node metastasis with the presence of SSTR2A in MTC [421]. In contrary to these observations, SSTR1 has been reported with a better response for MTC treatment [447]. Several previous studies have shown the use of radiolabeled SST analogs in treating differentiated and nondifferentiated thyroid tumours [429,446,449,450,451,452,453,454,455]. However, this is restricted due to the lack of precise and exact expression of the SSTR subtype, specifically in FTC and ATC. The SST analogue pasireotide, with an ability to target multiple SSTR subtypes, exhibits antiproliferative effects in MTC. Moreover, the use of pasireotide in combination with everolimus, an mTOR inhibitor, is proposed with improved therapeutic values in selective thyroid tumours [456]. It is also worth mentioning here that SSTR scintigraphy has been used to detect cerebellar metastasis of PTC [457]. Although controversies exist, the role of SST and its analogues in thyroid tumour diagnosis and prognosis cannot be denied in a receptor and tumour-specific manner.

## 28. Somatostatin, Somatostatin Receptors and Ovarian Tumour

### 28.1. Ovarian Cancer

Several previous studies have demonstrated that gynecological malignancy, including ovarian cancer, is the seventh most common tumour globally in women. Ovarian cancer is believed to be fatal and is the eighth reason of death all over the world. Furthermore, demographical distribution suggests a higher incidence of ovarian cancer in high-income countries when compared with lower- or middle-income countries. Patients at an early stage of ovarian cancer are often devoid of symptoms, suggesting that the tumour is at a quite advanced stage. Ovarian cancer is divided into many groups depending upon the type of cells of tumour origin. Three different types of cells, namely epithelial, stromal and germ cells, give rise most benign and malignant ovarian cancer [458]. In the origin of ovarian carcinoma, three possible sources include the surface of the ovary, epithelium of the fallopian tube and mesothelium-lined peritoneal cavity [459]. Most malignant ovarian cancers are of epithelial origin and represent a complex disease due to the morphological, biochemical, molecular and pathological variations, as well as multiple prognostic approaches [458,460,461,462,463]. Such cancers are different in many respects and divided into five different types, including high- and low-grade serous carcinoma, endometrioid, clear cell and mucinous carcinoma [464,465,466]. There are four different stages of ovarian cancer, stages I to IV. Stage I cancer resides in the ovarian tissue (one or both ovaries), whereas stage IV cancer which is the most advanced stage of tumor that also spreads to the lung and liver. In addition, stage II ovarian cancer invades the fallopian tubes, bowel and bladder, whereas stage III cancer metastasizes to lymph nodes and beyond the pelvis. Because of the lack of early reliable diagnosis of ovarian cancer, such different stages of ovarian cancer are determinant factors in success of treatment and survival time. Often malignant ovarian cancer metastasizes uniquely with less common hematogenous mode of spreading to the other body areas (Figure 6) and is believed to be the leading cause of death. Possible treatment avenues for ovarian cancer are the limited and frequent occurrence of resistance to chemotherapy resulted in failure of treatment that is the prominent cause of tumour progression and increasing rate in mortality. The family history of ovarian malignancy is the most prominent risk factor for the occurrence of ovarian cancer [467]. Increased risk of ovarian cancer at the molecular levels linked to the mutation of BRCA1 and -2 genes. Furthermore, environment and lifestyle also play a critical role in ovarian cancer, in addition to other gynecological problems. The mortality rate with ovarian cancer is relatively higher, and often resistance to chemotherapy is considered a major cause. Ernst Lengyel described ovarian cancer as a deadly disease that involves compressed visceral organs due to the aggressive growth of tumour cells which are not highly sensitive to chemotherapy [459]. It is also believed that ovarian cancer cells’ transformation is associated with genetic and epigenetic changes. During the process of cell division, epigenetic changes are preserved, and cells may acquire new functions associated with the tumor ability to spread to adjacent or distant tissues, as well as be resistant to treatment. Consistent with these observations, epigenetic therapies might serve as an effective treatment for ovarian cancer. With the significant progress in understanding the molecular details of tumor progression, what role tumor microenvironment in ovarian cancer might play in treatment strategies and preferences is worthy of investigation.

Because of the lack of an adequate screening procedure, tumour diagnosis is limited before the tumour invades other tissues. As a result, most ovarian cancers patients are diagnosed with disseminated disease and at an advanced stage that poses certain limitations, including lack of proper prognosis. Previous studies have demonstrated that paclitaxel in combination with platinum is a plausible treatment for ovarian cancer with a limited beneficial effect in the beginning. In contrast, long-term use of paclitaxel due to toxicities and resistance posed certain limitations [468]. Although beyond the scope of this review, additional limitations have also been identified, which are associated with developing resistance with the use of the paclitaxel treatment regime in ovarian cancer.

### 28.2. Role of Somatostatin and Somatostatin Receptor Expression in Ovarian Cancer

SST for its antiproliferation activity is known as a potential and effective therapeutic drug in tumour biology of neuroendocrine tumours but has not been explored in ovarian cancer. The use of SST and its receptor-specific agonists might play a crucial role in the suppression of resistance and in minimizing the toxicity often encountered with the use of paclitaxel treatment. The use of SST analogs as a potential therapeutic approach in ovarian cancer treatment is further supported by recent studies describing the inhibition of cell proliferation and apoptosis induction [468]. Furthermore, our recent observations on inhibition of EGF effect and dissociation of EGFR complex formation with the use of SST in breast cancer cells might serve as a possible mechanism to avert tumour growth resistance and toxicities as proposed in the use of cisplatin [468,469,470]. In ovarian cancer, the overexpression of SSTR2 supports its role in opposing resistance to paclitaxel treatment. Interestingly, conjugate prepared with SST analogs OCT has shown better outcome in tumour growth blockade even in case of tumour which exhibits a low expression of SSTR subtype. Most importantly, Chen Xi et al., using A2780 Taxol ovarian tumour xenograft, demonstrated the role of the paclitaxel-OCT conjugate in reversing resistance to paclitaxel [470]. However, a detailed, comprehensive distribution of SSTR subtypes in many control and ovarian cancer tissues is still awaited. The presence of SSTR subtype in ovarian cancer tissues is most controversial [471,472]. Previous studies have shown the expression of SSTR subtypes in ovarian tumor tissues. Hall et al. described the expression of SST and SSTR subtypes in malignant epithelium, blood vessels and stromal cells and proposed that presence of SSTRs in malignant epithelium as an effective target for SST analogs [473]. Schultz et al. analyzed 42 ovarian cancer tissues and provided selective distribution of SSTR subtypes in a receptor-specific manner [474]. Furthermore, Sugiyama et al. also described the overexpression of SSTR in ovarian cancer in addition to the previously identified two other cases from different laboratories [475]. Taken together, from these previous observations, it is plausible to predict that SSTR’s expression in ovarian cancer can be used as a potential target to recognize ovarian cancer by imaging and encountering paclitaxel resistance, which is the only possible therapeutic approach with potential toxicities that can also be minimized by using SST analogs conjugate.

## 29. Somatostatin, Somatostatin Receptors and Gastrointestinal Tumour

### 29.1. Gastrointestinal Tumour

The gastrointestinal tract, or GI tract, is the main digestive system of the human body. Several critical organs, including the stomach, pancreas and intestines, are highly susceptible to developing lesions. A lesion that is generally recognized during other surgical processes constitutes a common endocrine tumour that accounts for 50–70% of GI tumours, which further include intestine, stomach, colon, rectum and appendix. GI tract cancers are among the highest incidence and rate of cancer-related deaths worldwide [476,477]. The last two decades are witness to the increasing incidence of GI tract tumours. Tumours of GI origin have a slow growth rate compared to other types but can also be aggressive. These tumours often stay asymptomatic for an extended time and could be functional or nonfunctional.

### 29.2. Somatostatin and Somatostatin Receptors in Gastrointestinal Tumours

SST is mainly secreted from mucosal delta cells and acts as a paracrine transmitter to inhibit the release of gut hormones, the absorption of nutrients, the mobility of the gut and emptying of the stomach [478,479]. All five SSTRs have been identified throughout the GI tract with various densities in a tissue-specific manner [5]. Most GI tract tumours express SSTR2 in a high frequency, followed by SSTR1, -5 and -3, with SSTR4 being the rare case [374]. An early study of 34 patients with metastatic GI tumours responded poorly to SST analog, with a stable disease as the best outcome observed only in the minority [480]. In the human stomach, all five SSTRs have been identified. SSTR2 and -5 displayed strong expression in three gastric carcinoma models, whereas SSTR3 showed weak or variable expression [481]. A high methylation level of the SST promoter region and silencing of SST transcript has been observed in gastric tumour samples, potentially associated with gastric carcinogenesis [482]. Moreover, it has been reported that SST and its analogs inhibit the growth of gastric cancer cell lines [483,484], indicating a positive role of SST in regulation of gastric tumour progression. In clinical studies, SST and its analogs have proven to be effective on anti-angiogenesis and pro-necrosis and apoptosis in gastric cancer patients [485,486].

### 29.3. Therapeutic Approach and SST in Treatment of Gastrointestinal Tumours

It is believed that NET is genetically independent and restricted to the localization of the tumour in the target. Gastroenteropancreatic neuroendocrine tumour (GEP NET) is a comparatively rare and complex neoplasm. Depending on the site of the origin and whether the tumour is functionally active, clinical symptoms of GEP NET may be distinct and hard to diagnose. In addition to surgical resection as a potential treatment of GI tumours, the administration of receptor peptide-based radiolabelled compound is preferred. Amongst them, intravenous administration of SST analogs and SSTR agonists serve as a potential therapeutic avenue to improve quality of life with extended survival. They have been the most effective treatment of GI tumours.

GEP NETs have a high expression level of SSTR2 and -5, and a lower level for SSTR1, -3 and -4 [487]. This feature renders SST and its analogs an effective diagnosis method and the mainstay of symptomatic treatment [488,489]. SST analogs, however, are not so effective in controlling tumour size: only 5% of patients have tumour regression while 40–60% of patients show tumour stabilization [490,491,492]. Earlier, clinical trials with the use of OCT, LAR and lanreotide have provided promising results, with significant improvement in the median time to progression in patients diagnosed with metastatic midgut NETs and prolongation of progression-free survival in patients with hormonally nonfunctioning GEP NETs [493].

## 30. Colorectal Carcinoma and Somatostatin

Annual cases of colorectal cancer (CRC) diagnosed worldwide are almost a million and constitute the third most prevalent and highly malignant tumor affecting males and females at the age of 50 and above [356,494,495]. CRC is known to be aggressive and have a poor prognosis. It is classified in two different pathological types: small cell carcinoma and moderately differentiated type of tumor [496].

### 30.1. Expression of Somatostatin and Somatostatin Receptors in Colon Cancer

CRC express SSTR1 most, followed by SSTR5 and SSTR2 [497]. SST and its analogs have been shown to inhibit tumour growth in colorectal cancer cell lines and models positive to SSTR2 both in vivo and in vitro. SST has also been shown to mediate antiproliferative effects through SSTR3 and -5 in different CRC cell lines [498]. Furthermore, SST has been shown to control the feedback system between colon neuroendocrine cells and colon cancer stem cells through SSTR1 to regulate stem cell quiescence and proliferation [499]. It has been suggested that SSTRs regulate Na+/H+ exchanger in a cAMP-independent manner in colon carcinoma cells. In normal human colon and cancer tissues, comparable distribution of SST is reported in addition to reduced expression in cancer tissue when compared to control [499,500,501]. Studies also showed that poorly differentiated CRC exhibits a lower level of SST expression; whether SST is associated with metastasis of cancer is not established yet. Several previous studies have demonstrated the expression of SSTR1 subtypes in control and tumor colon tissues at the level of protein and mRNA with a distinct pattern of distribution [57,497,499,502]. However, sparsely distributed studies support a relation between the expression level of SSTR1 subtypes and tumor stage and lymph node metastasis [497]. The relative expression of SSTR2 is also variable in different control and tumor tissues but observed frequently [57,497,502,503,504,505]. Interestingly, SSTR2 expression is associated with tumour stage, type, localization and most importantly, with the presence of carcinoembryonic antigen in serum [497]. Furthermore, higher-grade tumor tissues were devoid of SSTR2 expression [57]. Moreover, SSTR2 mRNA has also been associated with cancer related death [504]. In comparison, the SSTR3 and -4 expression level is normally detected low in both control and tumor tissues and both receptors expression is linked to the stage of tumor [497]. SSTR5 was found to be highly expressed in control and tumor tissue often with high and frequent expression in colorectal cancer in association with decreased expression with tumor stage [497,502]. Also, no relation of SSTR5 with tumor stage is reported. Reubi, et al. showed that high expression of SSTR subtypes in veins in tumor vicinity and proposed role in metastasis mechanism for CRC [506]. In addition to normal and human tumor tissues variable expression of SST and SSTR subtypes has been described in in vitro in cultured colon cancer cells. SST and SSTR1-5 are well expressed in HT29 cells derived from low grade differentiated tumor whereas cell obtained from high grade and undifferentiated colon cancer namely SW480 are devoid of SST and SSTR5 [499]. Furthermore SSTR3, -4 and -5; SSTR3 and -5; and SSTR2, -3 and -5 are also expressed in HT-29, Caco-2 and HCT119 cancer cells, respectively [498]. Taken into consideration, SSTR subtypes are well expressed in normal and colorectal tissues in vivo and in vitro in tumor cultured cells.

### 30.2. Role of Somatostatin in Treatment of Colorectal Carcinoids

Studies have shown that SST is not only a therapeutic alternative in colon cancer treatment but has also been used as a tool in detection of colon cancer with high expression of SSTR subtypes. Hypermethylation of the SST promoter region has also been associated with uncontrolled cell proliferation in colorectal cancer [501]. So far in clinical studies, the effect of SST and its analogs on preventing tumour progression in patients with colorectal cancers are not any different than a placebo [507,508].

The type and stage of colon cancer are critical in the determination of treatment, and growing evidence supports the use of surgery, chemotherapy, radiotherapy and targeted therapies either alone or in combination. Schmoll et al. showed that, in the case of stage III and high-risk stage II patients with colon cancer, adjuvant treatment should begin with 8–12 weeks post-resection of tumor [509]. The existing literature supporting the use of SST analogues and receptor-specific analogues in diagnosis and prognosis is limited. Kostenich et al. showed the use of fluorescent SST conjugate in diagnosis of colon cancer [495]. The use of SST and its cognate receptors SSTR1-5 are well recognized not only in the diagnosis of a variety of tumors but also an effective therapeutic intervention of various tumors of distinct origin, including CRC. However, the mechanism for the role of SST involving in regulation of colon cancer management is not well understood due to complex contributing factors, including molecular and environmental in the development of CRC. The significant morbidity, including dehydration, electrolyte imbalance, renal failure and malnutrition is mostly seen in CRC due to high output of stoma, a life-threatening clinically relevant complication. Keeping high stroma in mind, inhibition of gastrointestinal secretion with SST analogues is an indication that SST might be an effective therapeutic alternative in colon cancer treatment [510,511,512,513]. It is believed that loss of SST synthesis in mucosa of patients with colon cancer might contribute to enhanced cell proliferation and promoter hypermethylation, possibly responsible for epigenetically reduced expression of SST in CRC. Furthermore, cell death and suppression of cell proliferation upon treatment of colon cancer cells with OCT support the role of SST in regulation of epithelial cell kinetics [501]. Although the molecular mechanisms involved are not well understood, the excess population of neoplastic stem cells exerts a crucial role in colorectal cancer initiation and progression [499]. SSTR1-positive cells in close proximity to colonic stem cells have been reported, but what role SSTR1 might play in the overproduction of neoplastic stem cells is not known and needs future attention. Increased expression of gastrin receptors and COX-2 is clinically observed in most CRC patients. SST via activation of SSTR3 and -5 decreases COX2 expression and resulted in inhibition of cell proliferation of colon cancer cells through inhibition of MAPK pathways via PTP activation [498].

## 31. Exploring the Concept of GPCRs Dimerization in Pathophysiology of Cancer: Synthesis of Chimeric Molecules

In addition to the several genetic mutations in tumours, the activation or suppression of cell surface receptor proteins plays a crucial and determinant role in most tumour cases. Such genetic mutation-specific and selective types, as well as the origin of the tumour, have been associated with the diagnosis and prognosis of the disease. By gain and/or loss of function, such genetic mutations also dictate the suppression or progression of the tumour via the modulation of certain downstream signaling pathways and determine the failure or success of the therapeutic regime in a tumour-specific manner. Amongst several receptors, members of the receptor tyrosine kinase family, including epidermal growth factor receptors, also commonly known as ErbBs (ErbB1-4), are predominantly expressed in large numbers of different types of tumours and served as crucial experimental tools in prognosis and diagnosis. It is now well established and undisputed that ErbBs in different types of tumours exhibited overexpression and function as a homo- and heterodimer within the family. There is ample evidence that EGFR1 (ErbB1) functionally interacts with ErbB2 and is linked to tumour growth, especially in breast cancer. Furthermore, ErbB2, once overexpressed in a tumour, exhibited homodimerization and resulted in tumour growth and often failure in treatment. The second class of cell surface receptor proteins with a potential role in tumour biology includes GPCRs, one of the most prominent families of seven helical transmembrane proteins. At present, GPCRs have been an effective target of >50% therapeutic drugs in several peripheral and central pathological conditions. Many tumours of different origins often exhibit the overexpression, as well as loss, of GPCRs, suggesting that this class of receptor protein plays a decisive role in tumour growth. Amongst them, SSTR subtypes that are members of the class A GPCR family are prominently expressed in most tumours and serve as a histopathological marker of disease. Moreover, SSTR subtypes are effective therapeutic interventions in suppressing tumour growth and a practical treatment choice. The co-expression of GPCRs within the family and other related members assures the possible crosstalk and formation of receptor complex distinct from their original entity in respect to their functional, physiological and pharmacological response to the cells. In the last two decades, significant progress has been made to explore the implication of protein–protein interaction, specifically GPCRs. However, the significance of such an interaction in pathological conditions is at a primitive stage and largely elusive. SSTR subtypes not only exhibit homo- and heterodimerization that is limited within the family but are also explored with other prominent members of the GPCR family, including dopamine, cannabinoid, opioid and adrenergic receptors [51,52,213,349]. The overexpression of SSTR1 and SSTR4 in MDA-MB-435S breast cancer cells resulted in a decrease in number of cells in the S phase upon receptor activation and associated with cell cycle arrest through dimerization between SSTR1/SSTR4 [200]. Interestingly, SSTR subtypes’ functional crosstalk is not restricted to GPCRs but has also been predicted with members of the RTKs family. Previous studies have shown that such interaction is not for the association but accounts for the dissociation of ErbBs that resulted in the suppression of signaling pathways associated with tumour progression [213]. The exploration of such interaction is promising in developing a new therapeutic strategy in cancer biology as described for the role of SSTR and dopamine receptors in the treatment of pituitary tumours [287,288]. The functional significance of crosstalk between SSTR and DR subtypes is discussed in details undersection pituitary tumor. Opioid receptors are also well expressed in different tumour cells and often co-expressed in a single cell [514]. Whether opioid receptors exhibit heterodimerization with SSTR subtypes or any other member of the GPCR family in cancer cells in vitro or in vivo is unknown. A comprehensive and systematic demonstration of GPCRs homo- and heterodimerization in tumour cells in vitro or in vivo holds specific promises. In addition, recent studies have shown SSTR, chemokine receptor 4 (CXCR4) and endothelin A receptor expression in grade I-IV brain tumours [515]. Furthermore, studies have also shown the use of SSTR2 and dopamine receptor 2 in combination with mTOR inhibitor in lung cancer cells NCI-H727 [382]. In addition to SSTR subtypes, several immunohistochemical studies showed the distribution of CXCR4 in pancreatic adenocarcinoma [259]. However, these observations revealed no therapeutic merit for SSTR and CXCR4 in pancreatic adenocarcinoma but proposed pathological significance indirectly through their effect on tumour microvessels [259]. Furthermore, specific emphasis on whether dimerization is constitutive of ligand associated will shed new light in association with receptor expression intensity. Future studies in this direction will help in designing new chimeric molecules that might serve better and safe therapy in tumours of different origins with variable genotypes.

## 32. Conclusions and Future Perspectives

From the discussion above, we can see that tumours represent one of the most complicated and heterogeneous pathologies because of several well-integrated signaling pathways and multiple genetic abnormalities in a tissue- and tumour-specific manner. Targeting single molecules is not a possible treatment, and the drugs working on multiple different avenues might serve as a better treatment to minimize the risk of tumour progression. In this direction, SST is the single peptide which has been proven as an effective treatment for different type of tumors via five different receptor subtypes directly or indirectly. Therefore, these tumours are ideal candidates for combination therapies which target multiple sites. Future studies should be directed to synthesize chimeric molecules along with chemotherapeutic strategies to block tumour proliferation, if not for complete eradication. The accumulated wealth of literature in the last five decades with significant contributions from different parts of the world attest to a road map of SST and cancer biology. The results discussed in this review govern several folds of significances and promises regarding the role of SST and its analogues in tumours of different origins and suggest that SST can be used as the best experimental tool in appropriate diagnosis and effective prognosis of most if not all tumours. Hopefully, the development of a new receptor-specific agonist with an ability to reach the tumour site, specifically in brain tumours, will continue promoting the use of SST as one of the most effective and efficient therapeutic interventions in tumour biology. The results from different studies in different types of tumours discussed here clearly advocate the role of SST in the regulation of multiple events associated with tumour progression.

## Figures and Tables

**Figure 1 ijms-25-00436-f001:**
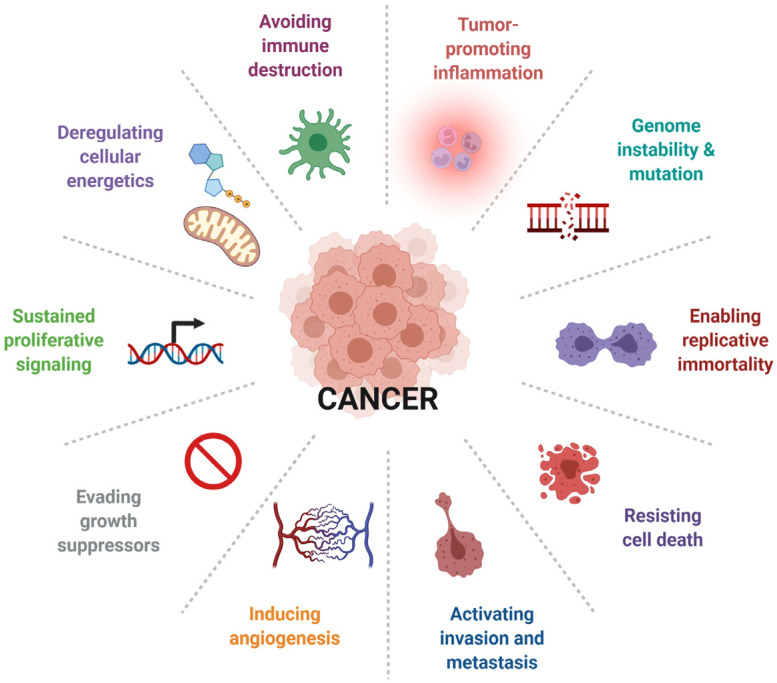
Schematic illustration showing multiple factors involved in tumor progression and treatment failure. Figure was created with BioRender.com.

**Figure 2 ijms-25-00436-f002:**
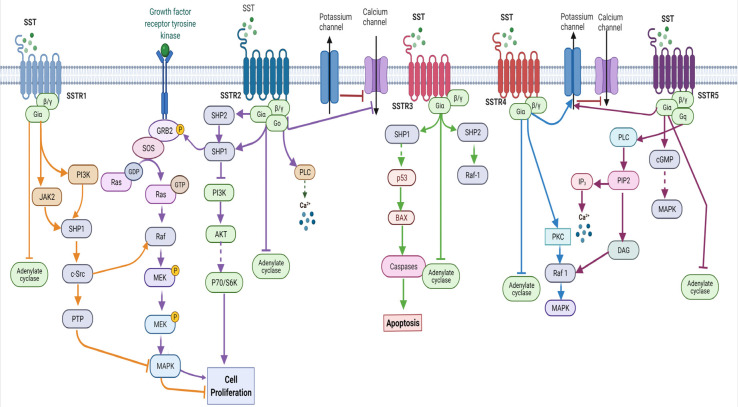
Schematic illustration showing somatostatin receptor-mediated signaling pathways. Figure was created with BioRender.com.

**Figure 3 ijms-25-00436-f003:**
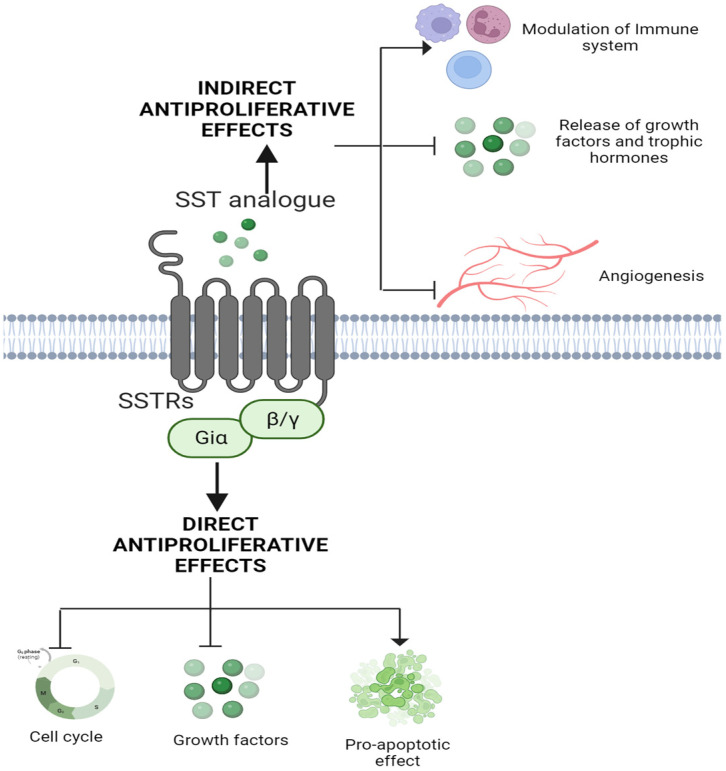
Schematic representation showing direct and indirect effect of somatostatin associated with inhibition of cell proliferation. Figure was created with BioRender.com.

**Figure 4 ijms-25-00436-f004:**
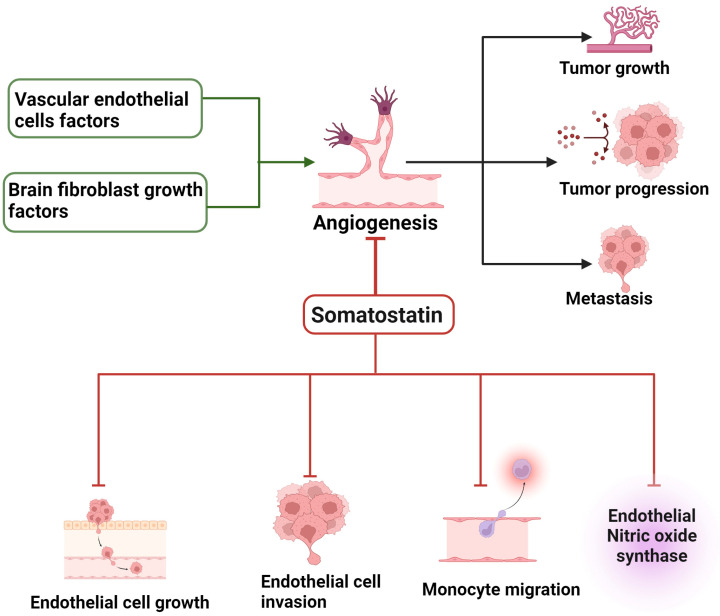
Schematic illustration displaying the process of sprouting angiogenesis and inhibitory role of SST in blood vessels. Figure was created with BioRender.com.

**Figure 5 ijms-25-00436-f005:**
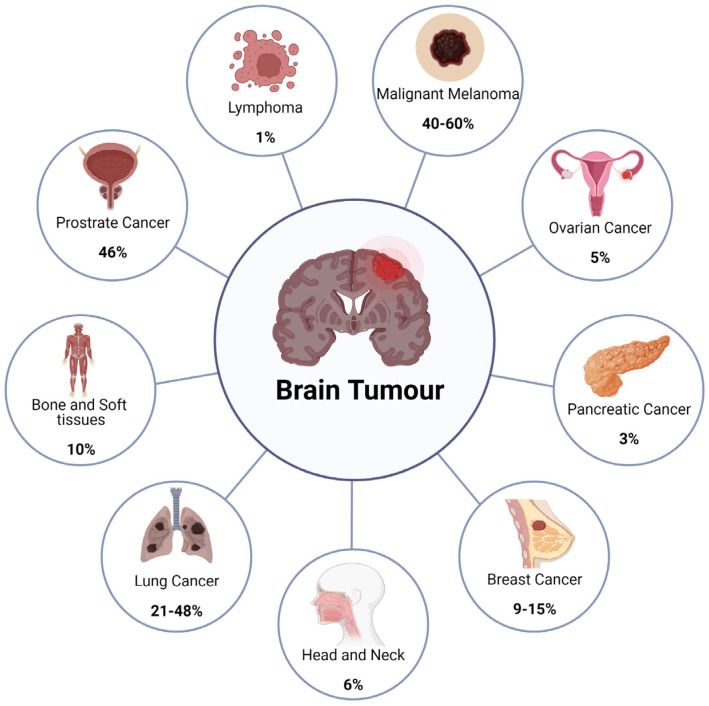
Different types of tumors that metastasize to the brain. Figure was created with BioRender.com.

**Figure 6 ijms-25-00436-f006:**
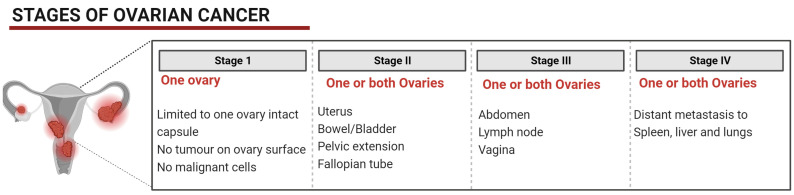
Schematic illustration depicting possible site of ovarian tumor in body. Figure was created with BioRender.com.

**Table 1 ijms-25-00436-t001:** List of SST analogues used in tumor treatment and their target site.

Somatostatin Analogues	Type of Cancer	Receptor Subtype Affinity
Octreotide (Sandostatin)	Breast cancer, prostrate cancer, gastrointestinal cancer, neuroendocrine tumors, exocrine pancreatic cancer, colorectal and hepatic cancer	SSTR2, SSTR5
Lanreotide (BIM23014)	Endocrine cancers, pituitary tumors, lung cancer, gut carcinoid, prostrate cancer, paraganglioma, pituitary adenoma, pheochromocytoma, meningioma	SSTR2, SSTR5
Pasireotide (SOM 230)	Breast cancer, lung cancer, colorectal cancer, hepatic cancer, endocrine cancers, gastrointestinal cancer	SSTR1–3, SSTR5
Vapreotide (RC-160)	Breast cancer, pancreatic cancer, lung cancer, ovarian cancer	SSTR2, SSTR5
Seglitide (MK 678)	GH-producing adenoma, gut carcinoid	SSTR2
CH-275	Prostrate and colorectal cancer	SSTR1
TT2–32	Prostrate and colon cancer	SSTR1
BIM23056	Nonfunctioning pituitary adenoma	SSTR3
BIM23066	GH-producing adenoma, gastrointestinal cancer, neuroblastoma, medulloblastoma, pheochromocytoma	SSTR2
Somatoprim (DG3173)	Pituitary adenomas	SSTR2, SSTR4, SSTR5
KE108	Pancreatic cancer	All SSTRs

## Data Availability

Not applicable.

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
