# Peer review of "Somatostatin and Somatostatin Receptors in Tumour Biology"

_ijms, 2023, doi:10.3390/ijms25010436_

Round 1

Reviewer 1 Report

Comments and Suggestions for Authors

This review is an extensive revisitation of the known effects of somatostatin and its receptors in in human tumors as main regulators of cell proliferation. Although several reviews have been already present in literature on somatostatin physiology and potential pharmacological use, to my knowledge, no one addressed the topic in such a complete manner. However, the enthusiasm for this study is somehow dampened by a rather diffuse disorganization of the article, when the role of somatostatin receptors in specific tumor histotypes is detailed.

Thus, my recommendation is to have a general reorganization of the review as detailed below, to allow the publication.

1)      There are several paragraphs, in which is just summarized what already described, ot what is going to be described in the following paragraphs See for example: paragraphs 5 and 9. The article is already extremely long; thus, I suggest avoiding repetition and delete these paragraphs (if necessary integrating them with the sections in which the topics are first discussed).

2)      The paragraphs in which clinical features of the tumors’ analyzed are really too long and out of scope with respect of the topic of the review (this is particularly evident for the “brain tumor” section). Moreover, these descriptions too much in detail in some cases contain some imprecisions (for ex. meningioma ARE NOT gliomas! Actually, they should be discussed in a separate paragraph. Similarly, gastrointestinal tumors, such as gastric and colorectal carcinomas are completely different nosologic entities respect to GEP-NET, and should discussed separately), and in others are completely unrelated to the topic, being not even discussed in relation of somatostatin receptors (for ex.par 15). Thus, I recommend to significantly reduce the general discussion of the tumor characteristic and limit those only to the ones related to somatostatin receptors.

3)      Paragraph 39 addresses a general aspect of the potential use of innovative somatostatin receptor agonist (i.e., chimeric molecules) as antiproliferative agents and the role of SSTR dimerization, should be moved before the tumor details.

4)      The sequence of the tumor discussed should also reorganized to follow a more logic order. In particular I would start with pituitary adenomas and NET, for which somatostatin analogue clinical use in already largely recognized. In these tumors. Moreover, preclinical and clinical evidence of antitumor of novel agonists and chimeric compounds should be introduced (for ex as far as GH- and TSH-secreting pituitary adenomas and NFPA (actually these adenomas are not mentioned and should be introduced, among the other SSTR-expressing pituitary tumors).

5)      Care should be played when studies are defined “recent”, since in several cases the Author refers to 20-year-old studies (see for ex. page 6 lines 6-8, ref 41, 42 and 43 are dated 2002,2003, and 2006; page 30, 8th line from the top, ref 384 is dated 2005).

6)      In other cases, citations in some paragraphs are not related to the tumor discussed: in par. 10 (page 11) dealing with brain tumors, refs 86 and 89 refer to breast and pancreas cancer, respectively. In par 18 (page 16) refs 26, 168 and 169 are reporting studies in CHO cells or pancreatic tumor cells. In this latter case there are several studies reporting phosphatase modulation by somatostatin analogues in glioma cells (just as an example you can see PMID 1856648, 15123617, 11223155). The last paper indicated (PMID 11223155) can also be referred when discussed SSTRs suppression of EGF phosphorylation in glioma, instead of (or along with) the long list of breast cancer studies (par 16 page 14). Several other imprecisions in the reference are present: for ex. pag 14 among the studies reporting OCT antiproliferative effects in glioma and medulloblastoma is cited ref 94, dealing only with the expression of SSTRs.

7)      In Par 7 the antimigratory properties of activated SSTRs should also be mentioned, being migration and invasion a prerequisite for neoangiogenesis (for ex see PMID 168288967, 12902325)

8)      Dealing with meningioma proliferation modulation by SSTR (page 15), I believe that the original evidence cited, in which an increase in proliferation by SSTRs is reported, were never repeated in several more recent studies (included one cited, but other are present in literature). Thus, the discussion of the antiproliferative role of cAMP seems a little bit out of context and, at least should be integrated with the observation that other intracellular mechanisms can be involved.

9)      Page 5: ref 21 seems to deal with SHP-2 and not SHP-1

Author Response

Response to Reviewer 1:

This review is an extensive revisitation of the known effects of somatostatin and its receptors in in human tumors as main regulators of cell proliferation. Although several reviews have been already present in literature on somatostatin physiology and potential pharmacological use, to my knowledge, no one has addressed the topic in such a complete manner. However, the enthusiasm for this study is somehow dampened by a rather diffuse disorganization of the article when the role of somatostatin receptors in specific tumor histotypes is detailed.

Author’s Response: I am very thankful to this reviewer for critical and thorough reading and valuable comments on my work. One of the most impactful components of reviewer critiques is dissecting the incorrect citation of references, and I am highly thankful to the reviewer for pointing out this ignorance. Reviewer comments helped me enormously in improving the quality of this review article.

I appreciate the reviewer's comments that “to my knowledge, no one addressed the topic in such a complete manner.”

  • There are several paragraphs which just summarise what was already described, or what is going to be discussed in the following paragraphs.
  • For example: paragraphs 5 and 9. The article is already extremely long; thus, I suggest avoiding repetition and delete these paragraphs (if necessary integrating them with the sections in which the topics are first discussed).

Author’s Response: I agree with the reviewer that a few sentences are repeated. For example, para 5 and 9 are written to address two different concepts. Importantly, these two paragraphs are written to articulate different segments of review for a smooth transition.

  • The paragraphs in which clinical features of the tumors’ analyzed are really too long and out of scope with respect of the topic of the review (this is particularly evident for the “brain tumor” section). Moreover, these descriptions too much in detail in some cases contain some imprecisions (for ex. meningioma ARE NOT gliomas! Actually, they should be discussed in a separate paragraph. Similarly, gastrointestinal tumors, such as gastric and colorectal carcinomas are completely different nosologic entities respect to GEP-NET, and should discussed separately), and in others are completely unrelated to the topic, being not even discussed in relation of somatostatin receptors (for ex. par 15). Thus, I recommend to significantly reduce the general discussion of the tumor characteristic and limit those only to the ones related to somatostatin receptors.

Author’s Response: I thank the reviewer for a constructive comment.

Somatostatin and brain tumors have not been studied as aggressively as other tumors. Therefore, I collected most of the information and put it together as a good reference.  

In this revised version, paragraph 15 (dealing with primary CNS lymphoma) was embedded intentionally to characterize that all CNS tumor are not always associated with SST despite the rich expression of SSTR subtypes.

All types of brain tumors are discussed separately, whether it is receptor distribution or treatment. I am aware that the receptor expression and treatment section should have been written in a tumor-respective manner, but I intentionally avoided a number of headings.

In this revised version, the section dealing with colorectal carcinomas (see pages 38 and 39) is extended separately for a detailed description.

Also, the last sentence in paragraph 11, stating that glioma is divided has been deleted.

  • Paragraph 39 addresses a general aspect of the potential use of innovative somatostatin receptor agonist (i.e., chimeric molecules) as antiproliferative agents and the role of SSTR dimerization, should be moved before the tumor details.

Author’s Response: Paragraph 39 is my conclusive message regarding the choice and preferences of tumor treatment. This concept was developed after our observation of D2R and SSTR5 heterodimerization. As suggested, this section can move, but it will be too early to discuss the concept of heterodimerization and chimeric molecules in between reviews.

4)      The sequence of the tumor discussed should also reorganized to follow a more logical order. In particular, I would start with pituitary adenomas and NET, for which somatostatin analogue clinical use is already largely recognized. In these tumors. Moreover, preclinical and clinical evidence of antitumor of novel agonists and chimeric compounds should be introduced (for ex as far as GH- and TSH-secreting pituitary adenomas and NFPA (actually these adenomas are not mentioned and should be introduced among the other SSTR-expressing pituitary tumors).

Author’s Response: In this revised version, the whole section defining the NFPA is included (please see pages 27 and 28). Existing publications on this subject are presented as the reviewer advised. I organize this review in a different manner for general readers as well as students engaged in SST and cancer biology. Most previous studies are presented as the reviewer suggested. However, I wanted to write this review a little differently than others. Therefore, I would like to request reviewer to approve this in the present organization.

5)      Care should be played when studies are defined “recent”, since in several cases the Author refers to 20-year-old studies (see for ex. page 6 lines 6-8, ref 41, 42 and 43 are dated 2002,2003, and 2006; page 30, 8th line from the top, ref 384 is dated 2005).

Author’s Response: This critical approach of the reviewer to review this work is highly appreciated, and I have corrected such a statement throughout this review. The word recent or recently has been changed accordingly.

6)      In other cases, citations in some paragraphs are not related to the tumor discussed: in par. 10 (page 11) dealing with brain tumors, refs 86 and 89 refer to breast and pancreas cancer, respectively. In par 18 (page 16) refs 26, 168 and 169 are reporting studies in CHO cells or pancreatic tumor cells. In this latter case there are several studies reporting phosphatase modulation by somatostatin analogues in glioma cells (just as an example you can see PMID 1856648, 15123617, 11223155). The last paper indicated (PMID 11223155) can also be referred to when discussing SSTRs suppression of EGF phosphorylation in glioma, instead of (or along with) the long list of breast cancer studies (par 16 page 14). Several other imprecisions in the reference are present: for ex. Page 14, among the studies reporting OCT antiproliferative effects in glioma and medulloblastoma, is cited in ref 94, dealing only with the expression of SSTRs.

Author’s Response: In this revised version, new references except one have been cited, and relevant information is added accordingly in this revised version of the review.

Inappropriate citations have either been deleted or corrected accordingly in this revised version.

7)      In Para 7 the antimigratory properties of activated SSTRs should also be mentioned, being migration and invasion a prerequisite for neoangiogenesis (for ex see PMID 16828967, 12902325)

Author’s Response: As suggested, additional references and observations have been added in paragraph 7 (see page 9); reviewer's suggestion is much appreciated.

8)      Dealing with meningioma proliferation modulation by SSTR (page 15), I believe that the original evidence cited, in which an increase in proliferation by SSTRs is reported, were never repeated in several more recent studies (included one cited, but other are present in literature). Thus, the discussion of the antiproliferative role of cAMP seems a little bit out of context and should at least be integrated with the observation that other intracellular mechanisms can be involved.

Author’s Response: I agree with the reviewer, and there is scanty literature on this subject. Keeping all these facts in mind, I definitely preferred to discuss certain unique studies that have not been supported extensively but exist in the literature. Just to provide an isolated example is the case of cAMP discussed here.

9)      Page 5: ref 21 seems to deal with SHP-2 and not SHP-1

Author’s Response: Thanks, this has been corrected in this revised resubmission

Reviewer 2 Report

Comments and Suggestions for Authors
  1. I have carefully read the attached paper. Here are three points that can further enhance the logical persuasion of the paper's argument:
    • Provide a more detailed explanation of the mechanism of action of somatostatin and somatostatin receptors.
    • Discuss the latest research trends on novel somatostatin analogues and specific receptor agonists.
    • Present additional research findings on the potential effects of somatostatin therapy in tumors of different origins.

Author Response

Response to Reviewer 2:

  1. Provide a more detailed explanation of the mechanism of action of somatostatin and somatostatin receptors.
  2. Discuss the latest research trends on novel somatostatin analogues and specific receptor agonists.
  3. Present additional research findings on the potential effects of somatostatin therapy in tumors of different origins.

Author’s Response: I am thankful to this reviewer for a very thoughtful comments on my work.  In this revised version, new literature is added, and mechanism of action for peptide are well described for each section of the present review

Round 2

Reviewer 1 Report

Comments and Suggestions for Authors

The Author addressed almost all my initial requests, but his justification to not change the overall strucutre of the manuscript is reasonable and acceptable.

I only recommend to check the reference list becaouse thera are still some very minor issues (below are listed only those I was able to detect but possibly others must be present)

1) ref 88 (Milano, M.T.; Zhang, H.; Metcalfe, S.K.; Muhs, A.G.; Okunieff, P. Oligometastatic breast cancer treated with curative-2201 intent stereotactic body radiation therapy. Breast Cancer Res Tr 2009, 115, 601-608) is cited in support of a sentence (...very little progress has been made to improve survival of brain tumour patients despite aggressive treatments because of significant rate of recurrence) dealing with brain tumors

2) ref 123 should be removed from the line 618 (only from there) since it is not dealing with cell proliferation

3) refs 331 and 333 are the same

4) reference to the lymphoma paragrah should be added

Author Response

Response to Reviewer

Author’s response: Firstly, I am very thankful to the reviewer for the time, effort and excellent review of this work. I have no doubt to say that this review would not have been the same without the reviewer's help. I am glad to know that the initial requests are well addressed and that the structure is acceptable.

Reference 88 is deleted, and Reference 123 is removed from line 618 but stays in the text.

Reference 333 is removed, and additional references in paragraphs 18 lymphoma are added as suggested.
